# Context-Based Dynamic Pricing with Partially Linear Demand Model

**Jinzhi Bu** [†]**, David Simchi-Levi** [‡§]**, Chonghuan Wang** [§]

[†] Department of Logistics and Maritime Studies, Faculty of Business,
The Hong Kong Polytechnic University
[‡] Institute for Data, Systems, and Society, Department of Civil and Environmental Engineering,
Operations Research Center, MIT
[§] Laboratory for Information and Decision Systems, MIT
jinzhi.bu@polyu.edu.hk, dslevi@mit.edu, chwang9@mit.edu

## Abstract

In today's data-rich environment, context-based dynamic pricing has gained much attention. To model the demand as a function of price and context, the existing literature either adopts a parametric model or a non-parametric model. The former is easier to implement but may suffer from model mis-specification, whereas the latter is more robust but does not leverage many structural properties of the underlying problem. This paper combines these two approaches by studying the context-based dynamic pricing with online learning, where the unknown expected demand admits a semi-parametric partially linear structure. Specifically, we consider two demand models, whose expected demand at price $p \in R^+$ and context $x \in \mathbb{R}^d$ is given by $bp + g(x)$ and $f(p) + a^\top x$ respectively. We assume that $g(x)$ is $\beta$-Hölder continuous in the first model, and $f(p)$ is $k$th-order smooth with an additional parameter $\delta$ in the second model. For both models, we design an efficient online learning algorithm with provable regret upper bounds, and establish matching lower bounds. This enables us to characterize the statistical complexity for the two learning models, whose optimal regret rates are $\widetilde{\Theta}(\sqrt{T} \vee T^{\frac{d}{d+2\beta}})$ and $\widetilde{\Theta}(\sqrt{T} \vee (\delta T^{k+1})^{\frac{1}{2k+1}})$ respectively. The numerical results demonstrate that our learning algorithms are more effective than benchmark algorithms, and also reveal the effects of parameters $d$, $\beta$ and $\delta$ on the algorithm's empirical regret, which are consistent with our theoretical findings.

## 1 Introduction

The recent success of online retailing has provided a data-rich environment for sellers to take better advantage of contextual information to dynamically improve their pricing strategies. Examples of such contextual information include products' characteristics, seasonality, festival information, economic indicators, etc. A fundamental question stemming from the presence of contexts is how to select a predictive model to capture the relationship among demand, price, and context.

In the current literature, there are two modeling approaches. The first approach is to adopt a parametric model, among which the linear model, i.e., demand = b × price + ⟨a, context⟩, is arguably the most fundamental one. The parametric assumption brings great convenience to practical implementation, and inspires many efficient learning algorithms. However, it may also lead to a serious issue of model mis-specification, as discussed in [25]. The second approach is to employ a fully non-parametric model, i.e., demand = $\mu$(price, context) for some unknown function $\mu(\cdot)$ without any parametric assumption. The non-parametric model can be quite robust to different application contexts. However,

36th Conference on Neural Information Processing Systems (NeurIPS 2022).

it does not leverage many structural properties of the underlying problem that may have been detected from huge amounts of historical data, resulting in much complexity for learning the function $\mu(\cdot)$.

Motivated by the idea of combining these two approaches, in this paper, we study context-based dynamic pricing with online learning where the unknown expected demand admits a semi-parametric partially linear structure. Such a structure preserves the advantages of simple interpretability from linear models and great flexibility from non-parametric models. Thus, it has also attracted significant attention in many other fields, especially when there exist high-dimensional contexts, such as statistics ([13]), econometrics ([36]), and biomedicines ([37]).

Formally, consider a seller selling products over a finite selling horizon of $T$ periods. At the beginning of each period $t$, the seller observes an independently and identically distributed (i.i.d.) context vector $x_t \in [0,1]^d \in \mathbb{R}^d$ encoding the product's characteristics and other exogenous information in period $t$. The seller then chooses a price $p_t$ from the feasible price range $[\underline{p}, \overline{p}] \subset (0, \infty)$ and customers observe the posted price $p_t$. The random demand $D_t$ is then generated according to the following function: $D_t(p_t) = \mu(p_t, x_t) + \varepsilon_t$, where $\mu(p_t, x_t)$ is an unknown function and $\{\varepsilon_t\}_{t \geq 1}$ is a sequence of i.i.d. sub-Gaussian random variables (r.v.'s) with zero mean and variance proxy $\sigma^2$, i.e., $\mathbb{E}[e^{\lambda \varepsilon_t}] \leq e^{\frac{\lambda^2 \sigma^2}{2}}$ for any $\lambda \in \mathbb{R}$. We consider two models for $\mu(p_t, x_t)$:

**Dynamic Pricing with Linear Pricing Effect (DPLPE):** $\mu(p_t, x_t) = bp_t + g(x_t)$ for unknown price elasticity $b < 0$ and $\beta$-Hölder continuous function $g : [0,1]^d \to \mathbb{R}$ with smoothness parameter $0 < \beta \leq 1$ (see Assumption 1);

**Dynamic Pricing with Linear Contextual Effect (DPLCE):** $\mu(p_t, x_t) = f(p_t) + a^\top x_t$ for unknown parameter $a \in \mathbb{R}^d$ and $k$th-order smooth function $f : [\underline{p}, \overline{p}] \to \mathbb{R}$ with degree $k \geq 1$ and parameter $\delta > 0$ that capture how tightly $f$ can be approximated by a polynomial function (see Assumption 2).

The seller observes demand $D_t$ and collects revenue $p_t D_t$. The expected revenue at price $p$ and context $x_t$ is denoted by $r(p, x_t) = p\mu(p, x_t)$ for any $p \in [\underline{p}, \overline{p}]$. An *admissible* pricing policy $\pi$ is defined as a sequence of functions $\{\pi_t(\cdot) : t \geq 1\}$, where each $\pi_t(\cdot)$ maps the historical information $(x_1, p_1, D_1, \ldots, x_{t-1}, p_{t-1}, D_{t-1}, x_t)$ observed up to period $t$ and possibly some external randomness to a feasible price in $[\underline{p}, \overline{p}]$. A clairvoyant optimal policy is assumed to know function $d(\cdot)$ and therefore always chooses the optimal price $p_t^* := \arg\max_{p \in [\underline{p}, \overline{p}]} r(p, x_t)$ in each period $t$. The performance of a pricing policy $\pi$ is measured by *regret* $R^\pi(T)$, defined as the difference between the $T$-period expected revenues generated by the clairvoyant optimal policy and by the pricing policy $\pi$. That is, $R^\pi(T) = \sum_{t=1}^{T} \mathbb{E}[r(p_t^*, x_t) - r(p_t, x_t)]$, where the expectation is taken over all the randomness associated with the environment and the algorithm.

## 1.1 Main results

Our main contributions lie in systematically characterizing the statistical complexity of the context-based dynamic pricing with partially linear demand models DPLPE and DPLCE. Throughout the paper, we use notations $\mathcal{O}(\cdot)$, $\Omega(\cdot)$ and $\Theta(\cdot)$ to hide constant factors, and use $\widetilde{\mathcal{O}}(\cdot)$, $\widetilde{\Omega}(\cdot)$ and $\widetilde{\Theta}(\cdot)$ to hide both constant and logarithmic factors.

**Statistical complexity for DPLPE.** For the first model DPLPE, we design an online learning algorithm that integrates the ideas of random shock pricing in [25] and contextual space binning and approximation. We prove that our algorithm achieves the theoretical regret upper bound $\widetilde{\mathcal{O}}(\sqrt{T} \vee T^{\frac{d}{d+2\beta}})$. We also establish the information-theoretic lower bound $\Omega(\sqrt{T} \vee T^{\frac{d}{d+2\beta}})$ for any admissible policy, which matches the regret upper bound up to logarithmic factors. The key novelty in our lower bound analysis is the construction of a family of high-dimensional $\beta$-Hölder continuous functions. To the best of our knowledge, this is also the first model that generalizes the commonly used Lipschitz continuous assumption to Hölder continuity in context-based dynamic pricing, with tight characterizations on the regret upper and lower bounds.

**Statistical complexity for DPLCE.** For the second model DPLCE, we devise a different learning algorithm based on the biased linear contextual bandit borrowed from [32] and our new idea of being more optimistic to chase the context-dependent optimal price. We prove that our algorithm achieves the regret upper bound $\widetilde{\mathcal{O}}(\sqrt{T} \vee (\delta T^{k+1})^{\frac{1}{2k+1}})$, and also establish a matching lower bound $\Omega(\sqrt{T} \vee (\delta T^{k+1})^{\frac{1}{2k+1}})$. Since $\delta$ is a parameter that depends on the specific problem instance $f(\cdot)$,

we refer to the regret bounds we derive as the *instance-dependent* bounds. To our knowledge, this is the first tight instance-dependent regret bound in context-based dynamic pricing with online learning. To achieve this, in the regret lower bound analysis, we need to construct a family of $k$th-order smooth functions that are adapted to parameters $k$ and $\delta$, which is also new to dynamic pricing literature.

## 1.2    Related work

**Non-contextual dynamic pricing.** There is a vast literature on dynamic pricing with online demand learning without context. These works can be classified into two categories: parametric models (see, e.g., [7], [18], [38]) and non-parametric models (see, e.g., [4], [5], [33],[32], [23]). We refer the reader to [12] for a comprehensive review on this stream of literature. Among these studies, [32] is the most relevant to our paper. [32] study dynamic pricing without context, and assume the expected demand is $k$th-order smooth in price for some integer $k$. They design a learning algorithm by applying the Optimism-in-the-Face-of-Uncertainty (OFU) principle borrowed from bandits, and prove the optimal regret is $\widetilde{\Theta}(T^{\frac{k+1}{2k+1}})$. The idea of algorithmic design in our DPLCE model is inspired by [32] but there are two major differences. First, the optimal price is stationary in [32], but can change over time in our setting. Therefore, the idea of multi-armed bandit protocol proposed in their paper for learning the stationary optimal price does not work. To overcome this difficulty, we propose a novel idea of being more optimistic over OFU in price selection to chase the context-dependent optimal price. Second, our notion of $k$th-order smoothness is more general than theirs by allowing $k$ not necessarily to be an integer and $\delta$ not necessarily to be a constant independent of $T$. The general values of $k$ and $\delta$ require a more careful selection of the parameters used by the algorithm, and a more sophisticated construction for the hard instances in the lower bound analysis.

**Context-based dynamic pricing.** There is a growing body of literature on context-based dynamic pricing (e.g., [27], [16], [22], [3], [24]). We refer to [3] for a recent review and briefly discuss the works related to this paper. [27] consider the demand model $a^\top x + bp$ in an incumbent-price setting, i.e., the expected demand for some given price is exactly known, and show that the greedy algorithm achieves the regret $\mathcal{O}(\log T)$. [25] study the same demand model as our DPLPE, and prove that the regret of a random price shock (RPS) algorithm compared with the clairvoyant who knows the best *linear* approximation to $g(x)$ is $\mathcal{O}(\sqrt{T})$. We borrow their RPS idea to design the learning algorithm for our DPLPE model, but analyze the regret of our algorithm relative to the optimal policy that knows the *true* demand model. [8] consider a non-parametric demand model in context-based dynamic pricing. For the Lipschitz continuous and locally concave revenue function, they prove that the optimal regret is $\widetilde{\Theta}(T^{\frac{d+2}{d+4}})$. This regret rate is strictly higher than $\widetilde{\Theta}(\sqrt{T} \vee T^{\frac{d}{d+2}})$ in our DPLPE model with $\beta = 1$ due to the separated effects of the price and context on demand we assume.

There is also a stream of works formulating demand using a binary choice model (see, e.g., [16], [21], [10], [30], [34], [14], [35], [20]). Every customer buys the product with probability $1 - F\left(p_t - \theta^\top x_t\right)$, where $F(\cdot)$ is the CDF of random noise. [14], [35] and [20] are the closest to our work. They also consider a semi-parametric form where $F(\cdot)$ is unknown non-parametric and $\theta$ is unknown parametric. Conceptually, one can transform our demand model into the one where each customer buys the product with probability $bp_t + g\left(x_t\right)$ (for DPLPE) or $f\left(p_t\right) + a^\top x_t$ (for DPLCE), assuming $bp_t + g\left(x_t\right)$ and $f\left(p_t\right) + a^\top x_t$ fall in $[0, 1]$. Since the modelling approaches are different, we are also facing different challenges. Different from the binary model, we need to estimate $b$ and the non-parametric function $g\left(\cdot\right)$ at the same time for DPLPE (or $a$ and the non-parametric function $f\left(\cdot\right)$ for DPLCE). Despite that we have an additive structure, it is still challenging to decouple $bp_t$ and $g\left(x_t\right)$ for DPLPE (or $f(p_t)$ and $a^\top x_t$ for DPLCE) with only a buying/not-buying feedback. In our model, the distribution of random shock makes no difference to the optimal pricing strategy. Thus, whether the distribution is parametric/known or not is not important in our formulation.

**Bandits with contextual information.** The most studied model in contextual bandit is the linear model (see, e.g., [2], [11], [29], [9], [1]), where the expected reward is a linear combination of contexts. The algorithms developed in these works are mostly built upon the celebrated idea of the OFU principle. In our DPLCE model, we borrow the OFU idea to design a learning algorithm. There are also a substantial amount of literature considering non-parametric reward feedback under Hölder continuous assumption (see, e.g., [28], [31], [26], [15]). Among these studies, [31] assuming a continuous action space is the most relevant to this work. For the general Lipschitz reward function, [31] proves that the optimal regret is $\widetilde{\Theta}(T^{\frac{d+d_p+1}{d+d_p+2}})$, where $d$ and $d_p$ are the dimensions for the context

space and action space respectively. Letting $d_p = 1$ leads to the optimal regret rate $\widetilde{\Theta}(T^{\frac{d+2}{d+3}})$ for a dynamic pricing problem. This rate is strictly higher than the optimal regret $\widetilde{\Theta}(\sqrt{T} \vee T^{\frac{d}{d+2}})$ in our DPLPE model with $\beta = 1$, and higher than $\widetilde{\Theta}(T^{\frac{2}{3}})$ in our DPLCE with $k = 1$ and $\delta = \Theta(1)$.

Table 1 provides a summary for the main results of this work and the mostly related literature.

Table 1: Summary of the main results in this work and existing results in literature

| Paper | Demand model | Regret upper bound | Regret lower bound | Key assumption |
|---|---|---|---|---|
| Dynamic pricing without context | | | | |
| [18] | $a + bp$ | $\widetilde{\mathcal{O}}(\sqrt{T})$ | $\Omega(\sqrt{T})$ | Linear demand model |
| [32] | $f(p)$ | $\widetilde{\mathcal{O}}(T^{\frac{k+1}{2k+1}})$ | $\Omega(T^{\frac{k+1}{2k+1}})$ | $f$: $k$th-order smooth |
| Dynamic pricing with context | | | | |
| [27] | $a^\top x + bp$ | $\mathcal{O}(\log T)$ | $\Omega(\log T)$ | Known incumbent price |
| [25] | $bp + g(x)$ | $\mathcal{O}(\sqrt{T})$ | $\Omega(\sqrt{T})$ | Benchmark: the best linear model |
| [31] | $d(p, x)$ | $\mathcal{O}(T^{\frac{d+2}{d+3}})$ | $\Omega(T^{\frac{d+2}{d+3}})$ | $d$: Lipschitz continuous |
| [8] | $d(p, x)$ | $\mathcal{O}((\log T)^2 T^{\frac{d+2}{d+4}})$ | $\Omega(T^{\frac{d+2}{d+4}})$ | $d$: Lipschitz continuous, smooth and locally concave revenue |
| [14] | $\mathbb{I}_{\{\theta^\top x + \epsilon \geq p\}}$ | $\widetilde{\mathcal{O}}(T^{\frac{2k+1}{4k-1}})$ | N.A. | $\epsilon$'s CDF: $k$th-order smooth ($k \geq 2$) |
| [35] | $\mathbb{I}_{\{\theta^\top x + \epsilon \geq p\}}$ | $\widetilde{\mathcal{O}}(T^{\frac{3}{4}})$ | $\Omega(T^{\frac{2}{3}})$ | No distributional assumptions |
| This work (Sec. 2 and Sec. 3) | $bp + g(x)$ | $\widetilde{\mathcal{O}}(\sqrt{T} \vee T^{\frac{d}{d+2\beta}})$ | $\Omega(\sqrt{T} \vee T^{\frac{d}{d+2\beta}})$ | $g$: $\beta$-Hölder continuous in $[0,1]^d$, |
| | $f(p) + a^\top x$ | $\widetilde{\mathcal{O}}(\sqrt{T} \vee (\delta T^{k+1})^{\frac{1}{2k+1}})$ | $\Omega(\sqrt{T} \vee (\delta T^{k+1})^{\frac{1}{2k+1}})$ | $f$: $k$th-order smooth with parameter $\delta$ |

## 2 Dynamic Pricing with Linear Pricing Effect (DPLPE)

In this section, we study the semi-parametric demand model with linear pricing effect, i.e., $\mu(p_t, x_t) = bp_t + g(x_t)$. We assume that the unknown price elasticity $b$ belongs to some known interval $[\underline{b}, \overline{b}] \subseteq (-\infty, 0)$, and the optimal price $p_t^* = -\frac{g(x_t)}{2b} \in [\underline{p}, \overline{p}]$ for any $b \in [\underline{b}, \overline{b}]$ and $x_t \in [0,1]^d$. We now formally introduce our assumption on function $g(\cdot)$.

**Assumption 1** *The function $g : \mathbb{R}^d \to \mathbb{R}$ is $\beta$-Hölder continuous for some parameter $0 < \beta \leq 1$, denoted by $g \in \mathcal{G}(\beta, d)$, meaning that there exists a constant $L > 0$, such that for any $x_1, x_2 \in [0,1]^d$, $|g(x_1) - g(x_2)| \leq L \|x_1 - x_2\|^\beta$.*

When $\beta = 1$, $\mathcal{G}(\beta, d)$ is the class of Lipschitz continuous functions, which is usually assumed in dynamic pricing literature (see, e.g., [8]). We consider a more general class of Hölder continuous functions with arbitrary $0 < \beta \leq 1$.

We first propose Algorithm 1 for our DPLPE model. Our algorithm combines two main ideas: (i) context space binning and approximation; and (ii) pricing with random shock.

**Context space binning and approximation.** To learn function $g(\cdot)$, the $\beta$-Hölder continuous assumption guarantees that it cannot change dramatically in a local area. This inspires a natural idea of dividing the context space $[0,1]^d$ into different small bins and using a constant to approximate $g(\cdot)$ in each bin. Specifically, we divide each coordinate of $[0,1]^d$ into $M$ equally sized segments and get $M^d$ small bins. We then estimate $\mathbb{E}[g(x)|x \in \mathbf{M}_j]$ (and $g(x_t)$ for any $x_t \in \mathbf{M}_j$) using the sample average approximation (SAA) from the residuals $\{d_s - p_s \hat{b}_t : 1 \leq s \leq t-1, x_s \in \mathbf{M}_j\}$, where $\hat{b}_t$ is an estimate for $b$ to be discussed in the next paragraph. It's worth noting that there exists an important trade-off for the choice of $M$. If $M$ is large, the amount of data collected for each bin will be limited, discouraging the success of learning $g(\cdot)$. If $M$ is small, we may incur a larger approximation error of $g(\cdot)$ by using a constant conditional expectation in each bin, leading to a suboptimal pricing strategy and a greater revenue loss. The specific choice of $M$ is given in Theorem 1.

**Pricing with random shock.** The idea of pricing with random shock is adopted from [25]. In each period $t$, the algorithm first computes a greedy price $p_t^g$ by plugging in the estimates for $g(x_t)$ and $b$, and then charges a price $p_t$ by adding a random shock $\Delta_t$, which takes the value of $\delta_t$ or $-\delta_t$ with equal probability, to the greedy price $p_t^g$. Following [25], instead of regressing $d_t$ against the true price $p_t$, we estimate $b$ by regressing $d_t$ against the price shock $\Delta_t$. Noting that $\frac{\sum_{s=1}^t \Delta_s d_s}{\sum_{s=1}^t \Delta_s^2} = \frac{\sum_{s=1}^t \Delta_s (bp_s + g(x_s) + \varepsilon_s)}{\sum_{s=1}^t \delta_s^2} = b + \frac{\sum_{s=1}^t \Delta_s (bp_s^g + g(x_s) + \varepsilon_s)}{\sum_{s=1}^t \delta_s^2}$. Since $\Delta_s$ is independent of

$bp_s^g + g(x_s) + \varepsilon_s$ for each $1 \le s \le t$ and has zero mean, $\frac{\sum_{s=1}^t \Delta_s d_s}{\sum_{s=1}^t \Delta_s^2}$ is an unbiased estimate for $b$. This also explains the necessity of regressing $d_t$ against the random shock $\Delta_t$.

---

**Algorithm 1:** Algorithm for Dynamic Pricing with Linear Price (`ADPLP`)

---

1 **Input:** price range $[\underline{p}, \overline{p}]$, bounds on the price coefficient $\underline{b}$ and $\overline{b}$, number of bins $M$

2 **Initialization:**

3 Partition each coordinate of $[0,1]^d$ into $M$ segments of equal length, denoted as $\mathbf{M}_j$ for $j = 1, 2, \cdots, M^d$;

4 Initialize $\mathcal{D}_{1,j} = \emptyset$ for each $j \in [M^d]$ and $\hat{b}_1 = \frac{\underline{b} + \overline{b}}{2}$;

5 **Main Steps:**

6 **for** $t = 1, 2, \cdots, T$ **do**

7     Set $\delta_t \leftarrow t^{-\frac{1}{4}}$;

8     Observe $x_t$ and find $j \in [M^d]$ such that $x_t \in \mathbf{M}_j$;

9     If $\mathcal{D}_{t,j} = \emptyset$, set $\hat{a}_{t,j} = -\hat{b}_t(\underline{p} + \overline{p})$; otherwise, set $\hat{a}_{t,j} \leftarrow \frac{\sum_{(x_k, p_k, d_k) \in \mathcal{D}_{t,j}} (d_k - p_k \hat{b}_t)}{|\mathcal{D}_{t,j}|}$ ;

10     Set unconstrained greedy price: $p_t^u \leftarrow -\frac{\hat{a}_{t,j}}{2\hat{b}_t}$;

11     Project greedy price: $p_t^g \leftarrow \mathbf{Proj}(p_t^u, [\underline{p} + \delta_t, \overline{p} - \delta_t])$;

12     Generate an independent random variable $\Delta_t \leftarrow \delta_t$ w.p. $\frac{1}{2}$ and $\Delta_t \leftarrow -\delta_t$;

13     Set price $p_t \leftarrow p_t^g + \Delta_t$;

14     Observe realized demand $d_t$;

15     Update $\mathcal{D}_{t+1,j} \leftarrow \mathcal{D}_{t,j} \cup \{(x_t, p_t, d_t)\}$ and $\mathcal{D}_{t+1,i} \leftarrow \mathcal{D}_{t,i}$ for $i \ne j$;

16     Update $\hat{b}_{t+1} \leftarrow \mathbf{Proj}(\frac{\sum_{s=1}^t \Delta_s d_s}{\sum_{s=1}^t (\Delta_s)^2}, [\underline{b}, \overline{b}])$;

17 **end for**

---

The following theorem shows the regret upper bound of Algorithm 1, whose proof is in Appendix A.

**Theorem 1** *Let Algorithm 1 run with $M = \lceil T^{\frac{1}{d+2\beta}} \rceil$. Under Assumption 1 and the known bound $[\underline{b}, \overline{b}]$ of the unknown $b$, the regret of Algorithm 1 is*

$$\widetilde{\mathcal{O}}\left(\sqrt{T} \vee T^{\frac{d}{d+2\beta}}\right).$$

The bound in Theorem 1 consists of two terms $\widetilde{\mathcal{O}}(\sqrt{T})$ and $\widetilde{\mathcal{O}}(T^{\frac{d}{d+2\beta}})$. The first term $\widetilde{\mathcal{O}}(\sqrt{T})$ arises from our introduction of random shocks reflecting the complexity of learning $b$. For the simplest linear demand model without context, the squared estimation error of $b$ incurred by an asymptotically optimal policy is in the order of $t^{-1/2}$ (see [18]). Therefore, a cumulative regret $\mathcal{O}(\sqrt{T})$ is in general unavoidable. The second term $\widetilde{\mathcal{O}}(T^{\frac{d}{d+2\beta}})$ is incurred by the inaccuracy of approximating the non-parametric function $g(\cdot)$ with a constant in each small bin. It captures the complexity for learning the non-parametric function $g(\cdot)$. This bound is monotonically increasing in dimension $d$, which agrees with the intuition that a higher dimension of the context space leads to a more challenging task of learning the function $g(\cdot)$. In addition, this bound decreases in $\beta$. This is consistent with the fact that if $g(\cdot)$ is smoother (i.e., $\beta$ is larger), it can be more accurately approximated by a constant in local bins. We also note that only when $d = 1$ and $\beta > \frac{1}{2}$ is $\widetilde{\mathcal{O}}(\sqrt{T})$ larger than $\widetilde{\mathcal{O}}(T^{\frac{d}{d+2\beta}})$. Otherwise, the regret upper bound is always $\widetilde{\mathcal{O}}(T^{\frac{d}{d+2\beta}})$.

We then establish the regret lower bound. We denote the regret of policy $\pi$ as $R_{g,b,\mathcal{P},\mathcal{D}}^\pi(T)$ when the demand function is $bp + g(x) + \varepsilon$, and the distributions of $x$ and $\varepsilon$ are $\mathcal{P}$ and $\mathcal{D}$ respectively. We use $\mathcal{E}(\sigma)$ to denote the class of sub-Gaussian distributions with variance proxy $\sigma^2$.

**Theorem 2** *For any $\mathcal{G}(\beta, d)$, $0 < \underline{b} \le \overline{b}$ and $\sigma \ge 0$, there exists a constant $K_1 > 0$ independent of $T$, such that for any admissible policy $\pi$,*

$$\sup_{\substack{g \in \mathcal{G}(\beta, d), \\ b \in [\underline{b}, \overline{b}], \mathcal{P}, \mathcal{D} \in \mathcal{E}(\sigma)}} R_{g,b,\mathcal{P},\mathcal{D}}^\pi(T) \ge K_1 \cdot \left(\sqrt{T} \vee T^{\frac{d}{d+2\beta}}\right).$$

Theorem 2 shows that the regret upper bound achieved by Algorithm 1 in Theorem 1 is unimprovable in terms of the dependency on the learning horizon $T$. The first lower bound $\Omega(\sqrt{T})$ can be directly obtained from the existing literature, e.g., [17]. To prove the second lower bound $\Omega(T^{\frac{d}{d+2\beta}})$, we construct a series of Hölder continuous functions in $[0,1]^d$ that are difficult to distinguish from each other. We then apply the Bretagnolle–Huber inequality (see [6]) and KL divergence arguments to bound the regret of any algorithm from below. See our construction and detailed proof in Appendix B.

We next compare our result with two relevant papers [25] and [8]. In [25], the same demand model $bp + g(x)$ is considered, but the performance of their learning algorithm is benchmarked with the optimal policy for the "best" linear demand function. In this case, they show that the optimal regret is $\Theta(\sqrt{T})$. By contrast, our regret notion is defined against the optimal policy for the *true* demand function, and thus, our optimal regret has an extra term $\widetilde{\Theta}(T^{\frac{d}{d+2\beta}})$ capturing the complexity of learning the function $g(\cdot)$. In [8], the authors prove the optimal regret $\widetilde{\Theta}(T^{\frac{d+2}{d+4}})$ for general Lipschitz continuous demand functions (in both context and price) assuming local concavity of the revenue function. When $\beta = 1$, the demand function we consider here is a special case of theirs. For this case, our optimal regret becomes $\widetilde{\Theta}(\sqrt{T} \vee T^{\frac{d}{d+2}})$, which is strictly lower than theirs. This reduction of the optimal regret benefits from the separated effects of the price and context assumed in our model. In terms of the algorithmic design, the algorithm in [8] divides both the price space and context space into bins, and treats the learning problem in each small bin as independent ones. In our setting, we leverage all the historical data to estimate the price sensitivity and achieve *information sharing* among different bins in the context space.

## 3 Dynamic Pricing with Linear Contextual Effect (DPLCE)

In this section, we study the semi-parametric demand model with linear contextual effect, i.e., $\mu(p_t, x_t) = f(p_t) + a^\top x_t$. We assume $||a|| \leq \overline{a}$ for some constant $\overline{a} > 0$. We now introduce our assumption on function $f(\cdot)$.

**Assumption 2** *The function $f : [\underline{p}, \overline{p}] \to \mathbb{R}$ is $k$th-order smooth with degree $k \geq 1$ and parameter $\delta$, denoted by $f \in \mathcal{F}^k([\underline{p}, \overline{p}]; \delta)$, meaning that $f$ is $\mathfrak{b}(k)$-times differentiable on $[\underline{p}, \overline{p}]$ and*

$$\left| f^{(\mathfrak{b}(k))}(p) - f^{(\mathfrak{b}(k))}(p') \right| \leq \delta \left| p - p' \right|^{k - \mathfrak{b}(k)}, \quad \forall p, p' \in [\underline{p}, \overline{p}], \tag{1}$$

*where $\mathfrak{b}(k) := \sup\{i \in \mathbb{N} : i < k\}$ is the maximal integer that is strictly less than $k$.*

Eq. (1) requires that the $\mathfrak{b}(k)$th-order derivative of $f(\cdot)$ is $(k - \mathfrak{b}(k))$-Hölder continuous. This assumption also implies that all the $i$th-order derivatives are bounded for $i \leq \mathfrak{b}(k)$ (i.e., $C_0 := \sup_{0 \leq i \leq \mathfrak{b}(k)} \sup_{p \in [\underline{p}, \overline{p}]} |f^{(i)}(p)| < \infty$). When $k$ is integer, i.e., $\mathfrak{b}(k) = k - 1$, Eq. (1) becomes the Lipschitz-continuous condition and Assumption 2 reduces to Assumption 1 in [32]. Different from [32] that sees $\delta$ as a constant, we allow the order of $\delta$ to depend on $T$, e.g., $\delta = \mathcal{O}(T^{-\lambda})$ for some $\lambda \geq 0$. As will be seen later, $\delta$ shows how tightly $f(p)$ can be approximated by a polynomial function, and will have impacts on both the regret upper bound and lower bound.

We propose Algorithm 2 for DPLCE. The key ideas are three-fold: (i) local polynomial approximation; (ii) biased linear contextual bandit and OFU; and (iii) optimism over OFU for price selection.

**Local polynomial approximation.** The idea of local approximation for smooth non-parametric functions through polynomial functions is borrowed from [32]. Specifically, we partition the price interval $[\underline{p}, \overline{p}]$ into $N$ segments of equal size, denoted as $\mathbf{I}_1, \cdots, \mathbf{I}_N$. For each price segment $\mathbf{I}_j := [a_j, b_j)$, we use a polynomial function of degree $\mathfrak{b}(k)$: $P_{\mathbf{I}_j}(p) := \sum_{i=0}^{\mathfrak{b}(k)} \frac{f^{(i)}(a_j)}{i!} (p - a_j)^i$, to locally approximate the true function $f(p)$. To learn the the coefficients of $P_{\mathbf{I}_j}(p)$, we apply the ridge regression method. By our assumption that $f^{(\mathfrak{b}(k))}(p)$ is $(k - \mathfrak{b}(k))$-Hölder continuous, it can be verified from Taylor expansion that $|f(p) - P_{\mathbf{I}_j}(p)| \leq |\frac{(p - a_j)^{\mathfrak{b}(k)}}{\mathfrak{b}(k)!}| \cdot |f^{(\mathfrak{b}(k))}(a_j') - f^{(\mathfrak{b}(k))}(a_j)| \leq \delta \cdot \frac{(\overline{p} - \underline{p})^k}{\mathfrak{b}(k)! \cdot N^k} = \mathcal{O}(\frac{\delta}{N^k})$, where $a_j' \in [a_j, p]$. Therefore, $\delta$ reflects how tightly $f$ can be approximated by a local polynomial function of degree $\mathfrak{b}(k)$. Moreover, the number of price segments $N$ needs to be carefully selected to balance the accuracy of local polynomial approximation and the learning efficiency for each price segment. The specific choice of $N$ is given in Theorem 3.

---
**Algorithm 2:** Algorithm for Dynamic Pricing with Linear Context (`ADPLC`)
---
1  **Input:** time horizon $T$, price range $[\underline{p}, \overline{p}]$, degree $k$, parameter $C_0$, context dimension $d$, upper bound $\bar{a}$, number of price segments $N$, error control terms $\Delta$, noise variance proxy $\sigma$
2  **Initialization:**
3  Partition $[\underline{p}, \overline{p}]$ into $N$ segments of equal length, denoted as $\mathbf{I}_i$ for $i = 1, 2, \cdots, N$;
4  Initialize for all $i \in [N]$: $\mathcal{D}_i = \emptyset$.
5  **Main Steps:**
6  **for** $t = 1, 2, \cdots, T$ **do**
7     Observe $x_t$;
8     **for** $i = 1, 2, \cdots, N$ **do**
9        $(\hat{\theta}_{t,i}, \hat{a}_{t,i}, \gamma_{t,i}, V_{t,i}) = \texttt{RLC}(\mathfrak{b}(k), d, \mathbf{I}_i, \mathcal{D}_i, \Delta, 1/T^2, C_0, \sigma, \bar{a})$;
10       $\hat{r}_{t,i} = \max_{p \in \mathbf{I}_i} p \times \left( \langle \hat{\theta}_{t,i}, \varphi(p) \rangle + \langle \hat{a}_{t,i}, x_t \rangle + \gamma_{t,i} \sqrt{\phi(p, x_t)^\top V_{t,i}^{-1} \phi(p, x_t)} + \Delta \right)$;
11       $\hat{p}_{t,i} = \arg\max_{p \in \mathbf{I}_i} p \times \left( \langle \hat{\theta}_{t,i}, \varphi(p) \rangle + \langle \hat{a}_{t,i}, x_t \rangle + \gamma_{t,i} \sqrt{\phi(p, x_t)^\top V_{t,i}^{-1} \phi(p, x_t)} + \Delta \right)$;
12    **end for**
13    Select $i_t = \arg\max_{i \leq N-1} \hat{r}_{t,i}$ and charge $p_t = \hat{p}_{t, i_t}$;
14    Observe realized demand $d_t$;
15    Update $\mathcal{D}_{i_t} \leftarrow \mathcal{D}_{i_t} \cup \{(x_t, p_t, d_t)\}$;
16 **end for**
---

---
**Algorithm 3:** Regression with Linear Context (`RLC`)
---
1  **Input:** polynomial degree $k$, context dimension $d$, domain $\mathbf{I} = [l, u]$, history $\mathcal{D}$, bias $\Delta$, probability $\epsilon$, smoothness parameter $C$, noise variance proxy $\sigma$, upper bound $\bar{a}$
2  Compute $\lambda = \frac{(u-l)^{2k}-1}{(u-l)^2-1} + d$;
3  Compute $\gamma = \sigma \sqrt{(d+k) \ln\left(\frac{d+k+|\mathcal{D}|}{d+k}\right) - 2\ln\epsilon} + \lambda^{\frac{1}{2}}(C^2 k + \bar{a}^2)^{\frac{1}{2}} + \Delta\sqrt{|\mathcal{D}|}$;
4  Compute $V = \lambda I_{(k+d)\times(k+d)} + \sum_{(x,p,d)\in\mathcal{D}} \phi(p,x)\phi(p,x)^\top$, where $\phi(p,x) = (\varphi(p)^\top, x^\top)^\top$ and $\varphi(p) = (1, (p-l), \cdots, (p-l)^{k-1})^\top$;
5  Compute Ridge estimate
   $(\hat{\theta}, \hat{a}) = \arg\min \sum_{(x,p,d)\in\mathcal{D}} (d - \langle \theta, \varphi(p) \rangle - \langle a, x \rangle)^2 + \lambda(\|\theta\|_2^2 + \|a\|_2^2)$;
6  **Output:** $\hat{\theta}, \hat{a}, \gamma, V$
---

**Biased linear contextual bandit and OFU.** To suggest a good price for each segment, we borrow the biased linear contextual bandit from [32]. Specifically, for each $t \geq 1$ and $p_t \in \mathbf{I}_j$, we can rewrite $D_t(p_t)$ as follows: $D_t(p_t) = P_{\mathbf{I}_j}(p_t) + a^\top x_t + \beta_t = \theta_j^\top \varphi(p_t) + a^\top x_t + \beta_t$, where $\beta_t := f(p_t) - P_{\mathbf{I}_j}(p_t) + \varepsilon_t$, $\varphi(p_t) := (1, (p_t - a_j), \cdots, (p_t - a_j)^{\mathfrak{b}(k)})$ and $\theta_j \in \mathbb{R}^{\mathfrak{b}(k)+1}$ whose $i$th element is $\frac{f^{(i-1)}(a_j)}{(i-1)!}$ for $1 \leq i \leq \mathfrak{b}(k) + 1$. This model is similar to the linear contextual bandit in the literature (see, e.g., [1], [9]), where the OFU idea that selects actions in an optimistic way is commonly adopted. The key difference is that $\beta_t$ contains a biased term that is not mean-zero and depends on the pricing decision. We adapt the idea of adding an additional term $\Delta$ when implementing the OFU principle to compute an optimistic price and optimistic revenue from [32].

**Optimism over OFU for price selection.** In the absence of context, the optimal price in [32] is *fixed*. To learn this fixed optimal price, [32] propose the idea of embedding the biased linear contextual bandit into a multi-armed bandit protocol that treats each price segment as an arm. The segment to which the fixed optimal price belongs is regarded as the "best" arm. By contrast, in our context-based setting, the optimal price depends on the random context revealed in each period and changes over time. Thus, our "best" arm in each period also changes, and the idea of multi-armed bandit protocol in [32] does not work. To overcome this challenge, we propose a new idea of optimism over OFU, which proceeds as follows. We first implement the OFU principle to each price segment $\mathbf{I}_i$ and compute an optimistic price $\hat{p}_{t,i}$ and optimistic revenue $\hat{r}_{t,i}$. Then we choose the most optimistic price from the $N$ candidate prices $\hat{p}_{t,1}, \ldots, \hat{p}_{t,N}$ that achieves the highest optimistic revenue.

The following theorem presents the regret upper bound for Algorithm 2, whose proof is in Appendix C.

**Theorem 3** *Let Algorithm 2 run with $N = \lceil \delta^{\frac{2}{2k+1}} T^{\frac{1}{2k+1}} \rceil + 1$ and $\Delta = \delta(\overline{p} - \underline{p})^k / N^k$. Under Assumption 2 and $||a|| \leq \overline{a}$, the regret of Algorithm 2 is*

$$\widetilde{\mathcal{O}}\left(d\big(\sqrt{T} \vee (\delta T^{k+1})^{\frac{1}{2k+1}}\big)\right).$$

First, when $\delta$ is a constant and $k$ is an integer, the regret upper bound becomes $\widetilde{\mathcal{O}}(dT^{\frac{k+1}{2k+1}})$, recovering Theorem 1 in [32] for the setting without contexts (i.e., $d = 1$ and $x$ is a constant). As $k$ becomes larger while keeping $\delta$ a constant, the upper bound decreases according to $\widetilde{\mathcal{O}}(dT^{\frac{k+1}{2k+1}})$. In the extreme case of $k = \infty$, the upper bound becomes $\widetilde{\mathcal{O}}(d\sqrt{T})$, which coincides with the optimal regret rate for the linear model $d(p, x) = bp + a^\top x$ (see, e.g., [3])[1]. Second, when $\delta$ is not a constant whose order can shrink as $T$ increases, Theorem 3 shows an *instance-dependent* bound which further differentiates our result from that in [32]. We can see that when $\delta = \Omega(T^{-\frac{1}{2}})$, the regret bound is $\widetilde{\mathcal{O}}(\delta^{\frac{1}{2k+1}} T^{\frac{k+1}{2k+1}})$, and when $\delta$ decreases to $\mathcal{O}(T^{-\frac{1}{2}})$, the regret bound is always $\widetilde{\mathcal{O}}(\sqrt{T})$. This shows that the magnitude of $\delta$ greatly affects the regret upper bound, and the smaller $\delta$ is, the lower the regret bound will be. This is because when $\delta$ becomes smaller, the approximation error using a local polynomial function $\mathcal{O}(\frac{\delta}{N^k})$ becomes smaller, leading to more accurate estimates for $f(\cdot)$. Since the polynomial function is differentiable infinitely many times, its regret is $\widetilde{\mathcal{O}}(\sqrt{T})$ from Section 3.4.1 in [32]. Therefore, as $\delta$ decreases, the regret upper bound decays to $\widetilde{\mathcal{O}}(\sqrt{T})$, which is attained when $\delta = \mathcal{O}(T^{-\frac{1}{2}})$. To prove Theorem 3, by the idea of optimism over OFU, we can upper bound the regret by the length of the confidence interval which can be controlled by the well-known concentration inequality in [1].

We then establish the regret lower bound. Similar to Theorem 1, we denote the regret of policy $\pi$ by $R^\pi_{f,a,\mathcal{P},\mathcal{D}}(T, k, \delta)$ when the demand function is $f(p) + a^\top x + \varepsilon$, where $f \in \mathcal{F}^k([\underline{p}, \overline{p}; \delta])$, and the distributions of $x$ and $\varepsilon$ are $\mathcal{P}$ and $\mathcal{D}$ respectively.

**Theorem 4** *For any $\mathcal{F}^k([\underline{p}, \overline{p}; \delta)$, $\overline{a} \geq 0$, and $\sigma \geq 0$, there exists a constant $K_2 > 0$ independent of $T$ and $\delta$, such that for any admissible policy $\pi$,*

$$\sup_{\substack{f \in \mathcal{F}^k([\underline{p}, \overline{p}];\delta), \\ \|a\| \leq \overline{a}, \mathcal{P}, \mathcal{D} \in \mathcal{E}(\sigma)}} R^\pi_{f,a,\mathcal{P},\mathcal{D}}(T, k, \delta) \geq K_2 \cdot \left(\sqrt{T} \vee \big(\delta T^{k+1}\big)^{\frac{1}{2k+1}}\right). \tag{2}$$

Theorem 4 shows that the regret upper bound achieved by Algorithm 2 in Theorem 3 is unimprovable in terms of its dependency on the learning horizon $T$ and parameter $\delta$. Note that the upper bound in Theorem 3 grows linearly in $d$, but the lower bound in Theorem 4 is independent of $d$. This is because in the lower bound analysis, we simply consider $a = \mathbf{0}$. We leave the problem of analyzing the more complicated dimension-dependent lower bound as future research. The first lower bound $\Omega(\sqrt{T})$ in Eq. (2) directly follows from the existing literature (e.g., [17]). The proof of the second lower bound $\Omega((\delta T^{k+1})^{\frac{1}{2k+1}})$ relies on constructing a series of $k$th-order smooth functions with parameter $\delta$ and applying Pinsker's inequality to bound the total variation of two probability measures through their KL divergence. See our construction and detailed proof in Appendix D.

## 4 Numerical Study

In this section, we conduct a numerical study to test the empirical performances of our algorithms. We measure the performance of a learning algorithm $\pi$ by the relative regret defined as follows: $\frac{\sum_{t=1}^T \mathbb{E}[p_t^* d(p_t^*, x_t) - p_t d(p_t, x_t)]}{\sum_{t=1}^T \mathbb{E}[p_t^* d(p_t^*, x_t)]} \times 100\%$. For both models, we compare our algorithms with the linear greedy algorithm that estimates the demand function by a linear function and myopically selects the optimal price that maximizes the proxy revenue in each period $t$. For our DPLPE model, we also compare our algorithm with the random price shock (RPS) algorithm proposed by [25]. For each instance, we repeat our experiments for 50 independent runs, and approximate the relative regret by the empirical relative regret averaged over 50 runs.

---

[1] Note that the linear model is a special case of DPLCE by letting $k = \infty$.

**Numerical results for DPLPE model.** For our DPLPE model with demand function $d(p, x_t) = bp + g(x_t)$, we set $b = -5$, $\varepsilon \sim \mathcal{N}(0, 5)$, $[\underline{b}, \overline{b}] = [-8, -3]$, $[\underline{p}, \overline{p}] = [1, 19]$, $x_t \sim \text{Uniform}([0, 1]^d)$, and consider the following form of function $g(x)$:

$$g(x) := \begin{cases} 100 \times (D(x, \partial[0, 1]^d))^\beta + 100 & \text{if } D(x, \partial[0, 1]^d) \leq \frac{1}{4}, \\ 100 \times (\frac{1}{4})^\beta - 300 \times (D(x, \partial[0, 1]^d) - \frac{1}{4})^\beta + 100 & \text{otherwise}, \end{cases}$$

where $D(x, \partial[0, 1]^d)$ denotes the Euclidean distance between $x$ and the boundary of $[0, 1]^d$. In Figure 1(a), we compare the empirical relative regret of our ADPLP with those of RPS algorithm and the linear greedy algorithm. The figure shows that our algorithm performs much better than both algorithms. In Figure 1(b), we investigate the effect of $d$ on the empirical relative regret of ADPLP by varying $d$ in the set $\{1, 3, 10\}$ while keeping $\beta = 1$. As shown in the figure, as $d$ decreases, the regret decreases, which is consistent with our theoretical results. In Figure 1(c), we test the effect of $\beta$ on the empirical relative regret of ADPLP by varying $\beta$ in the set $\{0.2, 0.4, 0.6, 1\}$ while keeping $d = 1$. When $\beta$ is relatively small, a slight increment of $\beta$, e.g., $\beta = 0.2$ to $\beta = 0.4$, will lead to significant speed-up of the convergence. The results of $\beta = 0.6$ and $\beta = 1$ are very close, which indicates that when $\beta$ is relatively large, increasing $\beta$ may only bring limited improvement.

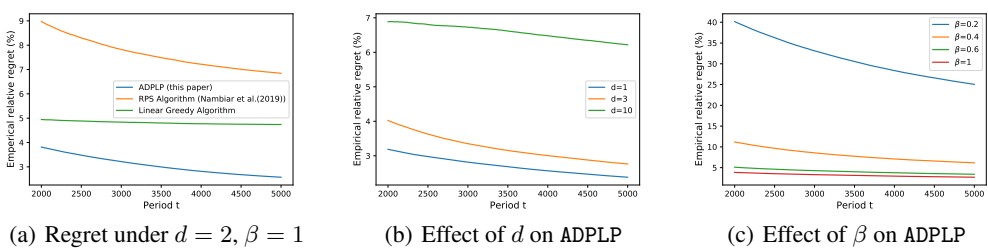

(a) Regret under $d = 2$, $\beta = 1$  (b) Effect of $d$ on ADPLP  (c) Effect of $\beta$ on ADPLP

Figure 1: Empirical relative regret for DPLPE model

**Numerical results for DPLCE model.** For our DPLCE model, we consider the following demand: $D_t(p_t) = -\frac{4}{15}\delta p_t^{2.5} + 30 + \frac{1}{d}\mathbf{1}_d \cdot x_t + \varepsilon_t$, where $\mathbf{1}_d$ denotes $(1, 1, \cdots, 1) \in \mathbb{R}^d$, $x_t$ is uniformly distributed on $[0, 1]^d$ independently and $\varepsilon_t \sim \mathcal{N}(0, 0.1^2)$. It can be verified that $k = 2.5$ in this case. In Figure 2(a), we fix $d = 2$, $\delta = 3.75$ and $[\underline{p}, \overline{p}] = [2.6, 3.8]$, and compare the empirical relative regret of our ADPLC with that of the linear greedy algorithm. The linear greedy algorithm has almost a constant empirical relative regret, showing that it fails to identify the optimal solution. By contrast, the empirical relative regret of ADPLC decreases as $t$ increases. We also test the effect of $\delta$ on the empirical relative regret of ADPLC by varying $\beta \in \{3.75, 2.5, 1\}$. As shown in Figure 2(b), as $\delta$ becomes smaller, the empirical relative regret decreases, which is consistent with our theoretic results.

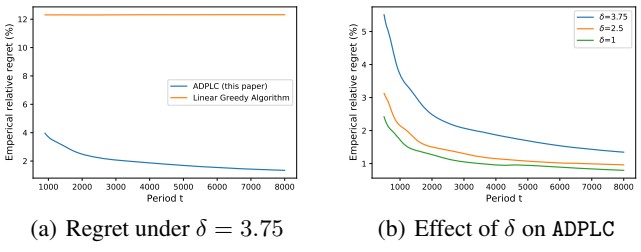

(a) Regret under $\delta = 3.75$  (b) Effect of $\delta$ on ADPLC

Figure 2: Empirical relative regret for DPLCE model

## 5   Conclusion

In this paper, we study the context-based dynamic pricing with online learning, where the unknown expected demand admits a partially linear semi-parametric structure. For both models DPLPE and DPLCE, we develop an efficient learning algorithm and characterize the optimal regret by proving matching upper and lower bounds. We note that our algorithms need to have the preliminary knowledge about the smoothness parameters $\beta$ in DPLPE model and $k$ and $\delta$ in DPLCE model.

One important direction for future research is to explore whether parameter-free algorithms can be developed and still achieve the optimal regret rate for both models.

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
