## Appendix A. Proof of Theorem 1

Here we provide a proof for Theorem 1. For each $t \in [T]$, let $j_t$ be the index of the bin that $x_t$ belongs. Conditional on $j_1, j_2, \ldots, j_T$, the total regret can be decomposed as follows:

$$\sum_{t=1}^{T} \mathbb{E}[r_t | j_1, j_2, \ldots, j_T]$$

$$= \sum_{t=1}^{T} \mathbb{E}\left[ \max_{p \in [\underline{p}, \overline{p}]} p(bp + g(x_t)) - p_t(bp_t + g(x_t)) \Big| j_1, j_2, \ldots, j_t \right]$$

$$= \sum_{t=1}^{T} \mathbb{E}\left[ -\frac{g(x_t)}{2b}\left( b \cdot \left( -\frac{g(x_t)}{2b} \right) + g(x_t) \right) - p_t\left( bp_t + g(x_t) \right) \Big| j_1, j_2, \ldots, j_t \right]$$

$$= |b| \sum_{t=1}^{T} \mathbb{E}\left[ \left( -\frac{g(x_t)}{2b} - p_t \right)^2 \Big| j_1, j_2, \ldots, j_t \right]$$

$$\leq 2|b| \sum_{t=1}^{T} \mathbb{E}\left[ \left( -\frac{g(x_t)}{2b} - p_t^0 \right)^2 + \left( p_t^0 - p_t \right)^2 \Big| j_1, j_2, \ldots, j_t \right]$$

$$\leq 2|b| \sum_{t=1}^{T} \mathbb{E}\left[ \left( -\frac{g(x_t)}{2b} - p_t^u \right)^2 + \left( p_t^0 - p_t \right)^2 \Big| j_1, j_2, \ldots, j_t \right]$$

$$\leq 6|b| \sum_{t=1}^{T} \mathbb{E}\left[ \underbrace{\left( \frac{-g(x_t) + \mathbb{E}[g(x)|x \in \mathbf{M}_{j_t}]}{2b} \right)^2}_{\text{regret from discretization error of } g(\cdot)} + \underbrace{\left( \frac{\mathbb{E}[g(x)|x \in \mathbf{M}_{j_t}]}{2b} - \frac{\mathbb{E}[g(x)|x \in \mathbf{M}_{j_t}]}{2\hat{b}_t} \right)^2}_{\text{regret from estimation error of } b} \right.$$

$$\left. + \underbrace{\left( \frac{\mathbb{E}[g(x)|x \in \mathbf{M}_{j_t}] - \hat{a}_{t,j_t}}{2\hat{b}_t} \right)^2}_{\text{regret from SAA}} + \underbrace{\left( p_t^0 - p_t \right)^2}_{\text{regret from random shock}} \Big| j_1, j_2, \ldots, j_t \right], \qquad (3)$$

where $p_t^0 := \text{Proj}(p_t^u, [\underline{p}, \overline{p}])$, the first identity holds since $x_t$ and $p_t$ are independent of $j_{t+1}, \ldots, j_T$, the first and the third inequalities follow from Cauchy-Schwarz inequality, and the second inequality is due to $-\frac{g(x_t)}{2b} \in [\underline{p}, \overline{p}]$.

The first term on the RHS of (3) arises from approximating $g(\cdot)$ using a constant in each local bin, and can be further upper bounded as follows due to Assumption 1:

$$\mathbb{E}\left[ \left( \frac{-g(x_t) + \mathbb{E}[g(x)|x \in \mathbf{M}_j]}{2b} \right)^2 \Big| j_1, \ldots, j_t \right] \leq \frac{\max_{x,y \in \mathbf{M}_j} (g(x) - g(y))^2}{4\underline{b}^2} \leq \frac{L^2 d^\beta}{4\underline{b}^2 M^{2\beta}}. \qquad (4)$$

The second term on the RHS of (3) is due to the estimation error of price sensitivity $b$ because

$$\mathbb{E}\left[ \left( \frac{\mathbb{E}[g(x)|x \in \mathbf{M}_j]}{2b} - \frac{\mathbb{E}[g(x)|x \in \mathbf{M}_j]}{2\hat{b}_t} \right)^2 \Big| j_1, \ldots, j_t \right]$$

$$\leq \frac{\max_{x \in [0,1]^d} (g(x))^2}{4\underline{b}^4} \mathbb{E}[(\hat{b}_t - b)^2 | j_1, \ldots, j_t]. \qquad (5)$$

The third term on the RHS of (3) represents the estimation error of $\mathbb{E}[g(x)|x \in \mathbf{M}_j]$ using the SAA method. We establish the following equation for this term whose proof is deferred to the end of this section:

$$\mathbb{E}\left[ \left( \frac{\mathbb{E}[g(x)|x \in \mathbf{M}_{j_t}] - \hat{a}_{t,j_t}}{2\hat{b}_t} \right)^2 \Big| j_1, \ldots, j_t \right] = 1_{\{|\mathcal{D}_{t,j_t}| \geq 1\}} \times \mathcal{O}\left( \mathbb{E}\left[ (b - \hat{b}_t)^2 | j_1, \ldots, j_t \right] + \frac{1}{|\mathcal{D}_{t,j_t}|} \right)$$

$$+ 1_{\{|\mathcal{D}_{t,j_t}| = 0\}} \times \mathcal{O}(1). \qquad (6)$$

The last term on the RHS of (3) comes from the regret of the random shock added to the greedy policy for exploration, and can be bounded by $\mathcal{O}(t^{-1/2})$:

$$\mathbb{E}[(p_t^0 - p_t)^2 | j_1, \ldots, j_T] \leq 2\mathbb{E}[(p_t^0 - p_t^g)^2 + (p_t^g - p_t)^2 | j_1, \ldots, j_T] \leq 4\delta_t^2 = \frac{4}{\sqrt{t}}. \qquad (7)$$

As seen from Eq. (5) and Eq. (6), the regret bound depends on the estimation error of $b$. To proceed, we establish the following upper bound on the estimation error of $b$, whose proof can be checked at the end of this section:

$$\mathbb{E}[(b - \hat{b}_t)^2 | j_1, \ldots, j_t] = \mathcal{O}\left(\frac{1}{\sqrt{t}}\right). \tag{8}$$

Putting Eq. (3) to Eq. (8) together and after some calculation, we can then establish the following upper bound on the total expected regret, whose detailed proof is also provided later in this section:

$$\sum_{t=1}^{T} \mathbb{E}[r_t] = \mathcal{O}\left(\frac{T}{M^{2\beta}}\right) + \mathcal{O}(\sqrt{T}) + \mathcal{O}(M^d \ln T). \tag{9}$$

By setting $M = \lceil T^{\frac{1}{d+2\beta}} \rceil$, we obtain the upper bound $\widetilde{\mathcal{O}}(\sqrt{T} + T^{\frac{d}{d+2\beta}}) = \widetilde{\mathcal{O}}(\sqrt{T} \vee T^{\frac{d}{d+2\beta}})$ in Theorem 1.

Now, it suffices to prove Eqs. (6), (8) and (9).

**Proof of Eq.** (6). Note that for each $t \geq 1$, when $|\mathcal{D}_{t,j_t}| \geq 1$, we have

$$\mathbb{E}\left[\left(\hat{a}_{t,j_t} - \mathbb{E}[g(x)|x \in \mathbf{M}_{j_t}]\right)^2 \Big| j_1, \ldots, j_t\right]$$

$$= \mathbb{E}\left[\left(\frac{1}{|\mathcal{D}_{t,j_t}|} \sum_{(x_k, p_k, d_k) \in \mathcal{D}_{t,j_t}} \left(d_k - p_k \hat{b}_t\right) - \mathbb{E}[g(x)|x \in \mathbf{M}_{j_t}]\right)^2 \Big| j_1, \ldots, j_t\right]$$

$$= \mathbb{E}\left[\left(\frac{1}{|\mathcal{D}_{t,j_t}|} \sum_{(x_k, p_k, d_k) \in \mathcal{D}_{t,j_t}} \left(bp_k + g(x_k) + \varepsilon_k - p_k \hat{b}_t\right) - \mathbb{E}[g(x)|x \in \mathbf{M}_{j_t}]\right)^2 \Big| j_1, \ldots, j_t\right]$$

$$\leq 3\mathbb{E}\left[\left(\frac{1}{|\mathcal{D}_{t,j_t}|} \sum_{(x_k, p_k, d_k) \in \mathcal{D}_{t,j_t}} p_k(b - \hat{b}_t)\right)^2 + \left(\frac{1}{|\mathcal{D}_{t,j_t}|} \sum_{(x_k, p_k, d_k) \in \mathcal{D}_{t,j_t}} \left(g(x_k) - \mathbb{E}[g(x)|x \in \mathbf{M}_{j_t}]\right)\right)^2\right.$$

$$\left. + \left(\frac{1}{|\mathcal{D}_{t,j_t}|} \sum_{(x_k, p_k, d_k) \in \mathcal{D}_{t,j_t}} \varepsilon_k\right)^2 \Big| j_1, \ldots, j_t\right]$$

$$\leq 3\mathbb{E}\left[\bar{p}^2(b - \hat{b}_t)^2 + \left(\frac{1}{|\mathcal{D}_{t,j_t}|} \sum_{(x_k, p_k, d_k) \in \mathcal{D}_{t,j_t}} \left(g(x_k) - \mathbb{E}[g(x)|x \in \mathbf{M}_{j_t}]\right)\right)^2 + \frac{\sigma^2}{|\mathcal{D}_{t,j_t}|} \Big| j_1, \ldots, j_t\right], \tag{10}$$

where the first identity follows from the definition of $\hat{a}_{t,j_t}$ and the inequality follows from Cauchy-Schwaz inequality.

We then bound the second term on the RHS of (10). Denoting the time indices when the feature vector falls into $\mathbf{M}_{j_t}$ by $1 \leq s_1 < s_2 < \ldots < s_{|\mathcal{D}_{t,j_t}|} \leq t - 1$, we have

$$
\mathbb{E}\Big[\Big(\frac{1}{|\mathcal{D}_{t,j_t}|} \sum_{(x_k,p_k,d_k)\in\mathcal{D}_{t,j_t}} \big(g(x_k) - \mathbb{E}[g(x)|x \in \mathbf{M}_{j_t}]\big)\Big)^2\Big|j_1,\ldots,j_t\Big]
$$

$$
= \mathbb{E}\Big[\Big(\frac{1}{|\mathcal{D}_{t,j_t}|} \sum_{k=1}^{|\mathcal{D}_{t,j_t}|} \big(g(x_{s_k}) - \mathbb{E}[g(x)|x \in \mathbf{M}_{j_t}]\big)\Big)^2\Big|x_{s_k} \in \mathbf{M}_{j_t}, \forall 1 \leq k \leq |\mathcal{D}_{t,j_t}|\Big]
$$

$$
= \mathrm{Var}\Big[\frac{1}{|\mathcal{D}_{t,j_t}|} \sum_{k=1}^{|\mathcal{D}_{t,j_t}|} \big(g(x_{s_k}) - \mathbb{E}[g(x)|x \in \mathbf{M}_{j_t}]\big)\Big|x_{s_k} \in \mathbf{M}_{j_t}, \forall 1 \leq k \leq |\mathcal{D}_{t,j_t}|\Big]
$$

$$
= \frac{1}{|\mathcal{D}_{t,j_t}|^2} \sum_{k=1}^{|\mathcal{D}_{t,j_t}|} \mathrm{Var}\Big[\big(g(x_k) - \mathbb{E}[g(x)|x \in \mathbf{M}_{j_t}]\big)\Big|x_{s_k} \in \mathbf{M}_{j_t}\Big]
$$

$$
= \frac{1}{|\mathcal{D}_{t,j_t}|} \mathrm{Var}\Big[g(y) - \mathbb{E}[g(x)|x \in \mathbf{M}_{j_t}]\Big|y \in \mathbf{M}_{j_t}\Big]
$$

$$
\leq \frac{1}{4|\mathcal{D}_{t,j_t}|} \max_{x,y\in\mathbf{M}_{j_t}} (g(x) - g(y))^2
$$

$$
\leq \frac{1}{4|\mathcal{D}_{t,j_t}|} \max_{x,y\in[0,1]^d} (g(x) - g(y))^2, \tag{11}
$$

where the second identity holds since conditional on $x_{s_k} \in \mathbf{M}_{j_t}$ for each $1 \leq k \leq |\mathcal{D}_{t,j_t}|$, $\frac{1}{|\mathcal{D}_{t,j_t}|} \sum_{k=1}^{|\mathcal{D}_{t,j_t}|} \big(g(x_{s_k}) - \mathbb{E}[g(x)|x \in \mathbf{M}_{j_t}]\big)$ is a mean zero random variable, the fourth identity holds since $x_{s_1}, x_{s_2}, \ldots, x_{s_{|\mathcal{D}_{t,j_t}|}}$ are i.i.d. random variables, and the first inequality follows from Hoeffding's lemma.

Therefore, we have the following upper bound on the third term of (3):

$$
\mathbb{E}\left[\left(\frac{\hat{a}_{t,j_t} - \mathbb{E}[g(x)|x \in \mathbf{M}_{j_t}]}{2\hat{b}_t}\right)^2\Big|j_1,\ldots,j_t\right]
$$

$$
= \mathbb{E}\left[1_{\{|\mathcal{D}_{t,j_t}|\geq 1\}} \times \left(\frac{\hat{a}_{t,j_t} - \mathbb{E}[g(x)|x \in \mathbf{M}_{j_t}]}{2\hat{b}_t}\right)^2\Big|j_1,\ldots,j_t\right] + \mathcal{O}(1) \times 1_{\{|\mathcal{D}_{t,j_t}|=0\}}
$$

$$
\leq 1_{\{|\mathcal{D}_{t,j_t}|\geq 1\}} \frac{3}{4\bar{b}^2} \mathbb{E}\left[\bar{p}^2(b - \hat{b}_t)^2 + \frac{\frac{1}{4}\max_{x,y\in[0,1]^d}(g(x) - g(y))^2 + \sigma^2}{|\mathcal{D}_{t,j_t}|}\Big|j_1,\ldots,j_t\right]
$$

$$
+ 1_{\{|\mathcal{D}_{t,j_t}|=0\}} \times \mathcal{O}(1), \tag{12}
$$

where the first identity holds since if $|\mathcal{D}_{t,j_t}| = 0$, a constant loss is incurred since $\hat{a}_{t,j_t}$, $\hat{b}_t$ and $\mathbb{E}[g(x)|x \in \mathbf{M}_{j_t}]$ are all bounded, and the inequality follows from (10) and (11) for $|\mathcal{D}_{t,j_t}| \geq 1$. This completes the proof of Eq. (6).

**Proof of Eq. (8).** We next bound $\mathbb{E}[(b - \hat{b}_t)^2 | j_1, \ldots, j_t]$ and prove Eq. (8). When $t \geq 2$, we have

$$
\mathbb{E}[(b - \hat{b}_t)^2 | j_1, \ldots, j_t] = \mathbb{E}\left[\left(\frac{\sum_{s=1}^t \Delta_s (bp_s^g + g(x_s) + \varepsilon_s)}{\sum_{s=1}^t \delta_s^2}\right)^2 \Big| j_1, \ldots, j_t\right]
$$

$$
= \frac{1}{(\sum_{s=1}^t s^{-\frac{1}{2}})^2} \mathbb{E}\left[\left(\sum_{s=1}^t \Delta_s (bp_s^g + g(x_s) + \varepsilon_s)\right)^2 \Big| j_1, \ldots, j_t\right]
$$

$$
= \frac{1}{(\sum_{s=1}^t s^{-\frac{1}{2}})^2} \mathbb{E}\left[\sum_{s=1}^t \Delta_s^2 (bp_s^g + g(x_s) + \varepsilon_s)^2 \Big| j_1, \ldots, j_t\right]
$$

$$
\leq \frac{3}{(\sum_{s=1}^t s^{-\frac{1}{2}})^2} \sum_{s=1}^t \mathbb{E}\left[\sum_{s=1}^t s^{-\frac{1}{2}}\left((bp_s^g)^2 + (g(x_s))^2 + \varepsilon_s^2\right) | j_1, \ldots, j_t\right]
$$

$$
\leq \frac{3\left(\underline{b}^2 \overline{p}^2 + \max_{x \in [0,1]^d}(g(x))^2 + \sigma^2\right)}{\sum_{s=1}^t s^{-\frac{1}{2}}}
$$

$$
\leq \frac{3\left(\underline{b}^2 \overline{p}^2 + \max_{x \in [0,1]^d}(g(x))^2 + \sigma^2\right)}{\sqrt{t}}, \tag{13}
$$

where the first identity follows from plugging in the definition of $\hat{b}_t$, the second identity holds since $\delta_s = s^{-\frac{1}{2}}$ by its definition, the third identity holds since when $s \neq k$, $\Delta_s$ is independent of $\Delta_k (bp_k^g + g(x_k) + \varepsilon_k)(bp_k^g + g(x_k) + \varepsilon_k)$, and from $\mathbb{E}[\Delta_s] = 0$, we have $\mathbb{E}[\Delta_s \Delta_k (bp_s^g + g(x_s) + \varepsilon_s)(bp_k^g + g(x_k) + \varepsilon_k)] = 0$, the first inequality follows from $\Delta_s \in \{-s^{-\frac{1}{4}}, s^{\frac{1}{4}}\}$ and Cauchy-Schwarz inequality, and the last inequality holds since when $t \geq 2$, $\sum_{s=1}^t s^{-\frac{1}{2}} \geq \int_1^{t+1} s^{-\frac{1}{s}} ds = 2(\sqrt{t+1} - 1) \geq \sqrt{t}$. This completes the proof of Eq. (8).

**Proof of Eq. (9).** Before we proceed to bound the total regret an prove Eq. (9), we note the following inequality on the summation for $1_{\{|\mathcal{D}_{t,j_t}| \geq 1\}} \frac{1}{|\mathcal{D}_{t,j_t}|}$:

$$
\sum_{t=1}^T 1_{\{|\mathcal{D}_{t,j_t}| \geq 1\}} \frac{1}{|\mathcal{D}_{t,j_t}|} = \sum_{j=1}^{M^d} \sum_{t=1}^T 1_{\{j_t = j, |\mathcal{D}_{t,j}| \geq 1\}} \frac{1}{|\mathcal{D}_{t,j}|} = \sum_{j=1}^{M^d} \sum_{k=1}^{|\mathcal{D}_{T,j}|} \frac{1}{k}
$$

$$
\leq \sum_{j=1}^{M^d} (1 + \log |\mathcal{D}_{T,j}|) = M^d + \log\left(\prod_{j=1}^{M^d} |\mathcal{D}_{T,j}|\right)
$$

$$
\leq M^d + \log\left(\frac{1}{M^d} \sum_{j=1}^{M^d} |\mathcal{D}_{T,j}|\right)^{M^d} \leq M^d\left(1 + \log T\right), \tag{14}
$$

where the second inequality holds since geometric mean is less than the arithmetic mean, and the last inequality holds since $\sum_{j=1}^{M^d} |\mathcal{D}_{T,j}| \leq T$.

Combining (12) and (14), we then obtain

$$
\sum_{t=1}^T \mathbb{E}\left[\left(\frac{\hat{a}_{t,j_t} - \mathbb{E}[g(x) | x \in \mathbf{M}_{j_t}]}{2\hat{b}_t}\right)^2 \Big| j_1, \ldots, j_t\right]
$$

$$
\leq \frac{3}{4\underline{b}^2} \sum_{t=1}^T \mathbb{E}\left[\overline{p}^2 (b - \hat{b}_t)^2 \Big| j_1, \ldots, j_t\right] + \left(\frac{1}{4} \max_{x,y \in [0,1]^d}(g(x) - g(y))^2 + \sigma^2\right) \sum_{t=1}^T 1_{\{|\mathcal{D}_{t,j_t}| \geq 1\}} \frac{1}{|\mathcal{D}_{t,j_t}|} + \mathcal{O}(M^d)
$$

$$
\leq \frac{3}{4\underline{b}^2} \sum_{t=1}^T \mathbb{E}\left[\overline{p}^2 (b - \hat{b}_t)^2 \Big| j_1, \ldots, j_t\right] + \left(\frac{1}{4} \max_{x,y \in [0,1]^d}(g(x) - g(y))^2 + \sigma^2\right) M^d(1 + \ln T) + \mathcal{O}(M^d),
$$

$$
\tag{15}
$$

where the first inequality follows from (12) and $\sum_{t=1}^T 1_{\{|\mathcal{D}_{t,j_t}| = 0\}} \leq M^d$, and the second inequality follows from (14).

Finally, putting Eqs. (3), (4), (7), (12), (13) and (15) together, we obtain the following upper bound on the total regret conditional on $j_1, \ldots, j_T$:

$$\sum_{t=1}^{T} \mathbb{E}[r_t | j_1, \ldots, j_T]$$

$$\leq 6|\underline{b}| \left( \frac{c_1 T}{M^{2\beta}} + c_2 \sum_{t=1}^{T} \mathbb{E}[(b - \hat{b}_t)^2 | j_1, \ldots, j_t] + 4 \sum_{t=1}^{T} t^{-\frac{1}{2}} + c_3 (1 + \ln T) M^d \right) + \mathcal{O}(M^d)$$

$$\leq 6|\underline{b}| \left( \frac{c_1 T}{M^{2\beta}} + c_4 \sum_{t=1}^{T} t^{-\frac{1}{2}} + c_3 (1 + \ln T) M^d \right) + \mathcal{O}(M^d)$$

$$\leq 6|\underline{b}| \left( \frac{c_1 T}{M^{2\beta}} + c_4 (2\sqrt{T} - 1) + c_3 (1 + \ln T) M^d \right) + \mathcal{O}(M^d), \tag{16}$$

where $c_1 = \frac{L^2 d^\beta}{4\underline{b}^2}$, $c_2 = \frac{1}{4\underline{b}^2} (\max_{x \in [0,1]^d} (g(x))^2 + 3\overline{p}^2)$, $c_3 = \frac{3}{4\underline{b}^2} (\frac{1}{4} \max_{x,y \in [0,1]^d} (g(x) - g(y))^2 + \sigma^2)$ and $c_4 = 3c_2 (\underline{b}^2 \overline{p}^2 + \max_{x \in [0,1]^d} (g(x))^2 + \sigma^2) + 4$. This completes the proof of Eq. (9).

By setting $M = \lceil T^{\frac{1}{d+2\beta}} \rceil$ and taking the expectation with respect to $j_1, \ldots, j_T$ on both sides of (16), we obtain the following upper bound on the total expected regret:

$$\sum_{t=1}^{T} \mathbb{E}[r_t] \leq 6|\underline{b}| \left( (c_1 + 2^d c_3 (1 + \ln T)) T^{\frac{d}{d+2\beta}} + c_4 (2\sqrt{T} - 1) \right) = \tilde{\mathcal{O}}(\sqrt{T} \vee T^{\frac{d}{d+2\beta}}),$$

which completes the proof of Theorem 1.

## Appendix B. Proof of Theorem 2

The first lower bound $\Omega(\sqrt{T})$ is directly implied from the existing results (e.g., Theorem 1 in [17]) by letting $g(x)$ be a constant function (which belongs to $\mathcal{G}(\beta, d)$). To show the second lower bound $\Omega(T^{\frac{d}{d+2\beta}})$, we first need to construct a series of Hölder continuous functions that are "similar" to each other and therefore difficult to distinguish. We partition the context space $[0,1]^d$ into $M^d$ equally sized bins, denoted as $\mathbf{M}_1, \mathbf{M}_2, \ldots, \mathbf{M}_{M^d}$, by dividing each dimension of the context space into $M$ intervals of equal length. We then construct a series of functions $g_{\mathbf{w}}(\cdot)$ indexed by a tuple $\mathbf{w} \in \{0,1\}^{M^d}$, and the $j$-th component of $\mathbf{w}$ determines the value of $g_{\mathbf{w}}(x)$ for $x \in \mathbf{M}_j$ as follows:

$$g_{\mathbf{w}}(x) = \begin{cases} |\underline{b}| (\underline{p} + \overline{p}) & w_j = 0 \\ |\underline{b}| (\underline{p} + \overline{p}) + (D(x, \partial \mathbf{M}_j))^\beta & w_j = 1 \text{ and } D(x, \partial \mathbf{M}_j) \leq \frac{1}{4^{1/\beta} M} \\ |\underline{b}| (\underline{p} + \overline{p}) + \frac{1}{4M^\beta} & w_j = 1 \text{ and } D(x, \partial \mathbf{M}_j) > \frac{1}{4^{1/\beta} M}, \end{cases} \tag{17}$$

where $\partial \mathbf{M}_j$ denotes the boundary of the bin $\mathbf{M}_j$, and $D(x, \mathbf{M}_j) := \inf\{\|x - y\| : y \in \partial \mathbf{M}_j\}$ denotes the Euclidean distance between $x$ and $\partial \mathbf{M}_j$. The following lemma shows that each $g_{\mathbf{w}}$ is Hölder continuous, whose proof is more sophisticated than [28] and [8], and deferred to Appendix B.1.

**Lemma 1** *For each $\mathbf{w} \in \{0,1\}^{M^d}$, $g_{\mathbf{w}}(\cdot)$ defined in (17) belongs to $\mathcal{G}(\beta, d)$ with $L = 2$.*

Let $\mathcal{P}$ be a uniform distribution on $\bigcup_{j=1}^{M^d} \{x \in \mathbf{M}_j : D(x, \partial \mathbf{M}_j) > \frac{1}{4^{1/\beta} M}\}$. By setting $b = \underline{b}$, whenever $x$ falls into $\mathbf{M}_j$, the optimal price $p^*(x)$ associated with any function $g_{\mathbf{w}}(\cdot)$ is

$$p^*(x) = \begin{cases} \frac{\underline{p} + \overline{p}}{2} & w_j = 0 \\ \frac{\underline{p} + \overline{p}}{2} + \frac{1}{8M^\beta |\underline{b}|} & w_j = 1. \end{cases} \tag{18}$$

We then consider two demand functions $g_{(\mathbf{w}_{-j}, w_j)}$ for $w_j = 0, 1$, where we use $(\mathbf{w}_{-j}, w_j)$ to denote an index $\mathbf{w} \in \{0,1\}^{M^d}$ whose $j$-th coordinate is $w_j$ and the other coordinates are $\mathbf{w}_{-j}$. Eq. (18) indicates that if the price charged by an algorithm in period $t$ is greater than $\frac{\underline{p} + \overline{p}}{2} + \frac{1}{16M^\beta |\underline{b}|}$, its gap with the optimal price under demand function $g_{(\mathbf{w}_{-j}, 0)}$ is greater than $\frac{1}{16M^\beta |\underline{b}|}$; and if the price

charged by the algorithm is less than $\frac{\underline{p}+\overline{p}}{2} + \frac{1}{16M^\beta|\underline{b}|}$, its gap with the optimal price under the other function $g_{(\mathbf{w}_{-j},1)}$ is still greater than $\frac{1}{16M^\beta|\underline{b}|}$. Bretagnolle–Huber inequality (see [6]) guarantees that the minimal error of making one type of the mistakes depends on how well the algorithm can distinguish between the two demand functions. This is formalized in the following inequality:

$$\mathbb{P}^{\pi,t}_{g_{(\mathbf{w}_{-j},0)}}\left(p_t \geq \frac{\underline{p}+\overline{p}}{2} + \frac{1}{16M^\beta|\underline{b}|}\,\bigg|\, x_t \in \mathbf{M}_j\right) + \mathbb{P}^{\pi,t}_{g_{(\mathbf{w}_{-j},1)}}\left(p_t < \frac{\underline{p}+\overline{p}}{2} + \frac{1}{16M^\beta|\underline{b}|}\,\bigg|\, x_t \in \mathbf{M}_j\right)$$

$$\geq \frac{1}{2}\exp\left(-\mathrm{KL}\left(\mathbb{P}^{\pi,t}_{g_{(\mathbf{w}_{-j},0)}}, \mathbb{P}^{\pi,t}_{g_{(\mathbf{w}_{-j},1)}}\right)\right), \tag{19}$$

where $\mathbb{P}^{\pi,t}_{g_{(\mathbf{w}_{-j},w_j)}}$ denotes the probability measure under policy $\pi$ up to period $t$ when the true demand function is $g_{(\mathbf{w}_{-j},w_j)}$. We then apply the chain rule of KL divergence to establish the following upper bound on the KL divergence between $\mathbb{P}^{\pi,t}_{g_{(\mathbf{w}_{-j},0)}}$ and $\mathbb{P}^{\pi,t}_{g_{(\mathbf{w}_{-j},1)}}$:

$$\mathrm{KL}(\mathbb{P}^{\pi,t}_{g_{(\mathbf{w}_{-j},0)}}, \mathbb{P}^{\pi,t}_{g_{(\mathbf{w}_{-j},1)}}) \leq \frac{1}{32\sigma^2 M^{d+2\beta}}t. \tag{20}$$

With a careful decomposition of the regret, by applying Eqs. (19) and (20) with $M = \lceil T^{\frac{1}{d+2\beta}}\rceil$, we will obtain the desired lower bound $\Omega(T^{\frac{d}{d+2\beta}})$.

In the subsequent analysis, we fix the price elasticity to be $\underline{b}$, the distribution of contexts to be a uniform distribution on $\bigcup_{j=1}^{M^d}\{x \in \mathbf{M}_j : D(x, \partial\mathbf{M}_j) > \frac{1}{4^{1/\beta}M}\}$, and the distribution of random noise to be normal distribution. For the ease of presentation, we omit the dependency of the regret and the expectation on these terms. For any policy $\pi$, we establish the following lower bound on its worst-case regret by restricting to the functions $g_{\mathbf{w}}(\cdot)$ constructed in (17):

$$\sup_{g\in\mathcal{G}(\beta,d)} R_g^\pi(T) \geq \sup_{g\in\{g_{\mathbf{w}}:\mathbf{w}\in\{0,1\}^{M^d}\}} R_g^\pi(T)$$

$$= |\underline{b}| \sup_{g\in\{g_{\mathbf{w}}:\mathbf{w}\in\{0,1\}^{M^d}\}} \sum_{t=1}^T \mathbb{E}_g^\pi\left[(p^*(x_t) - p_t)^2\right]$$

$$\geq \frac{|\underline{b}|}{2^{M^d}} \sum_{\mathbf{w}\in\{0,1\}^{M^d}} \sum_{t=1}^T \mathbb{E}_{g_{\mathbf{w}}}^\pi\left[(p^*(x_t) - p_t)^2\right]$$

$$\geq \frac{|\underline{b}|}{2^{M^d}} \sum_{\mathbf{w}\in\{0,1\}^{M^d}} \sum_{t=1}^T \sum_{j=1}^{M^d} \mathbb{E}_{g_{\mathbf{w}}}^\pi\left[(p^*(x_t) - p_t)^2 \mathbb{I}_{\{x_t\in\mathbf{M}_j\}}\right]$$

$$= \frac{|\underline{b}|}{2^{M^d}} \sum_{j=1}^{M^d} \sum_{\mathbf{w}_{-j}\in\{0,1\}^{M^d-1}} \sum_{w_j\in\{0,1\}} \sum_{t=1}^T \mathbb{E}_{g_{(\mathbf{w}_{-j},w_j)}}^\pi\left[(p^*(x_t) - p_t)^2 \mathbb{I}_{\{x_t\in\mathbf{M}_j\}}\right]$$

$$= \frac{|\underline{b}|}{2^{M^d}M^d} \sum_{j=1}^{M^d} \sum_{\mathbf{w}_{-j}\in\{0,1\}^{M^d-1}} \sum_{w_j\in\{0,1\}} \sum_{t=1}^T \mathbb{E}_{g_{(\mathbf{w}_{-j},w_j)}}^\pi\left[(p^*(x_t) - p_t)^2 \mid x_t \in \mathbf{M}_j\right], \tag{21}$$

where in the second and third identities, we use $(\mathbf{w}_{-j}, w_j)$ to denote an index $\mathbf{w}$ whose $j$-th coordinate is $w_j$ and the other coordinates are $\mathbf{w}_{-j}$, and in the third identity, we use the fact that $\mathbb{E}_{g_{\mathbf{w}}}^\pi[(p^*(x_t) - p_t)^2\mathbb{I}_{\{x_t\in\mathbf{M}_j\}}] = \mathbb{E}_{g_{\mathbf{w}}}^\pi[(p^*(x_t) - p_t)^2 \mid x_t \in \mathbf{M}_j]\mathcal{P}_u(x_t \in \mathbf{M}_j)$ and $\mathcal{P}_u(x_t \in \mathbf{M}_j) = \frac{1}{M^d}$ by our construction.

Noting the expression of the optimal price in Eq. (18), we have the following lower bound on $\mathbb{E}_{g_{(\mathbf{w}_{-j},w_j)}}^\pi[(p^*(x_t) - p_t)^2 \mid x_t \in \mathbf{M}_j]$ in the RHS of (21): for each $w_j \in \{0,1\}$,

$$\mathbb{E}_{g_{(\mathbf{w}_{-j},0)}}^\pi\left[(p^*(x_t) - p_t)^2 \,\bigg|\, x_t \in \mathbf{M}_j\right] \geq \left(\frac{1}{16M^\beta|\underline{b}|}\right)^2 \mathbb{P}^{\pi,t}_{g_{(\mathbf{w}_{-j},0)}}\left(\left\{p_t \geq \frac{\underline{p}+\overline{p}}{2} + \frac{1}{16M^\beta|\underline{b}|}\right\}\,\bigg|\, x_t \in \mathbf{M}_j\right), \tag{22}$$

$$\mathbb{E}^{\pi}_{g_{(\mathbf{w}_{-j},1)}}\left[(p^*(x_t) - p_t)^2 \,\Big|\, x_t \in \mathbf{M}_j\right] \geq \left(\frac{1}{16M^{\beta}\,|\underline{b}|}\right)^2 \mathbb{P}^{\pi,t}_{g_{(\mathbf{w}_{-j},1)}}\left(\left\{p_t \leq \frac{\underline{p}+\overline{p}}{2} + \frac{1}{16M^{\beta}\,|\underline{b}|}\right\}\,\Big|\, x_t \in \mathbf{M}_j\right),$$

$$(23)$$

where $\mathbb{P}^{\pi,t}_{g_{(\mathbf{w}_{-j},w_j)}}$ denotes the probability measure for $X_1, P_1, d_1, \ldots, X_t, P_t, d_t$ under policy $\pi$ and demand function $g_{(\mathbf{w}_{-j},w_j)}$.

Due to the above Eqs. (22) and (23) and Eq. (19), to further bound the RHS of (21), we next focus on analyzing the KL-divergence between the two probability measures $\mathbb{P}^{\pi,t}_{g_{(\mathbf{w}_{-j},0)}}(\cdot \mid x_t \in \mathbf{M}_j)$ and $\mathbb{P}^{\pi,t}_{g_{(\mathbf{w}_{-j},1)}}(\cdot \mid x_t \in \mathbf{M}_j)$, and prove Eq. (20). Noting the following identity

$$\mathbb{P}^{\pi,t}_{g_{(\mathbf{w}_{-j},w_j)}}(X_1, P_1, d_1, \cdots, X_t, P_t \mid x_t \in \mathbf{M}_j)$$
$$= \mathbb{P}^{\pi,t-1}_{g_{(\mathbf{w}_{-j},0)}}(X_1, P_1, d_1, \cdots, X_{t-1}, P_{t-1}, d_{t-1}) \times \pi(X_t, P_t \mid x_t \in \mathbf{M}_j, X_1, P_1, d_1, \cdots, X_{t-1}, P_{t-1}, d_{t-1}),$$

and denoting $\pi(X_t, P_t \mid x_t \in \mathbf{M}_j, X_1, P_1, d_1, \cdots, X_{t-1}, P_{t-1}, d_{t-1})$ as $\pi_t^j$, we obtain

$$\mathrm{KL}\big(\mathbb{P}^{\pi,t}_{g_{(\mathbf{w}_{-j},0)}}(\cdot \mid x_t \in \mathbf{M}_j), \mathbb{P}^{\pi}_{g_{(\mathbf{w}_{-j},1)}}(\cdot \mid x_t \in \mathbf{M}_j)\big)$$

$$= \mathrm{KL}(\mathbb{P}^{\pi,t-1}_{g_{(\mathbf{w}_{-j},0)}} \times \pi_t^j, \mathbb{P}^{\pi,t-1}_{g_{(\mathbf{w}_{-j},1)}} \times \pi_t^j)$$

$$= \mathrm{KL}(\mathbb{P}^{\pi,t-1}_{g_{(\mathbf{w}_{-j},0)}}, \mathbb{P}^{\pi,t-1}_{g_{(\mathbf{w}_{-j},1)}}) + \mathbb{E}^{\pi,t-1}_{g_{(\mathbf{w}_{-j},0)}}[\mathrm{KL}(\pi_t^j, \pi_t^j)]$$

$$= \mathrm{KL}(\mathbb{P}^{\pi,t-1}_{g_{(\mathbf{w}_{-j},0)}}, \mathbb{P}^{\pi,t-1}_{g_{(\mathbf{w}_{-j},1)}}), \qquad (24)$$

where the second identity follows from the chain rule of the KL divergence. Moreover, $\mathbb{P}^{\pi,t}_{g_{(\mathbf{w}_{-j},w_j)}}$ can be further decomposed as follows:

$$\mathbb{P}^{\pi,t-1}_{g_{(\mathbf{w}_{-j},w_j)}}(X_1, P_1, d_1, \cdots, X_{t-1}, P_{t-1}, d_{t-1})$$
$$= \mathbb{P}^{\pi,t-2}_{g_{(\mathbf{w}_{-j},w_j)}}(X_1, P_1, d_1, \cdots, X_{t-2}, P_{t-2}, d_{t-2}) \times \mathcal{P}_u(X_{t-1})$$
$$\times \pi(P_{t-1} \mid X_1, P_1, d_1, \cdots, X_{t-2}, P_{t-2}, d_{t-2}, X_{t-1}) \times \mu_{g_{(\mathbf{w}_{-j},w_j)}}(d_{t-1}|P_{t-1}, X_{t-1}). \quad (25)$$

Denoting $\pi(P_{t-1} \mid X_1, P_1, d_1, \cdots, X_{t-2}, P_{t-2}, d_{t-2}, X_{t-1})$ by $\pi_{\mathcal{F}_{t-2}, x_{t-1}}$, we get

$$\mathrm{KL}(\mathbb{P}^{\pi,t-1}_{g_{(\mathbf{w}_{-j},0)}}, \mathbb{P}^{\pi,t-1}_{g_{(\mathbf{w}_{-j},1)}})$$
$$= \mathrm{KL}(\mathbb{P}^{\pi,t-2}_{g_{(\mathbf{w}_{-j},0)}}, \mathbb{P}^{\pi,t-2}_{g_{(\mathbf{w}_{-j},1)}}) + \mathbb{E}^{\pi,t-2}_{g_{\mathbf{w}_{-j},0}}[\mathrm{KL}(\mathcal{P}_u \times \pi_{\mathcal{F}_{t-2}, x_{t-1}} \times \mu_{g_{(\mathbf{w}_{-j},0)}}, \mathcal{P}_u \times \pi_{\mathcal{F}_{t-2}, x_{t-1}} \times \mu_{g_{(\mathbf{w}_{-j},1)}})]$$
$$= \mathrm{KL}(\mathbb{P}^{\pi,t-2}_{g_{(\mathbf{w}_{-j},0)}}, \mathbb{P}^{\pi,t-2}_{g_{(\mathbf{w}_{-j},1)}}) + \mathbb{E}^{\pi,t-2}_{g_{(\mathbf{w}_{-j},0)}}[\mathrm{KL}(\mathcal{P}_u \times \pi_{\mathcal{F}_{t-2}, x_{t-1}}, \mathcal{P}_u \times \pi_{\mathcal{F}_{t-2}, x_{t-1}})]$$
$$+ \mathbb{E}^{\pi,t-2}_{g_{(\mathbf{w}_{-j},0)}}[\mathbb{E}_{\mathcal{P}_u \times \pi_{\mathcal{F}_{t-2}, x_{t-1}}}[\mathrm{KL}(\mu_{g_{(\mathbf{w}_{-j},0)}}, \mu_{g_{(\mathbf{w}_{-j},1)}})]]$$
$$= \mathrm{KL}(\mathbb{P}^{\pi,t-2}_{g_{(\mathbf{w}_{-j},0)}}, \mathbb{P}^{\pi,t-2}_{g_{(\mathbf{w}_{-j},1)}}) + \mathbb{E}^{\pi,t-2}_{g_{(\mathbf{w}_{-j},0)}}[\mathbb{E}_{\mathcal{P}_u \times \pi_{\mathcal{F}_{t-2}, x_{t-1}}}[\mathrm{KL}(\mu_{g_{(\mathbf{w}_{-j},0)}}, \mu_{g_{(\mathbf{w}_{-j},1)}})]], \qquad (26)$$

where the first and second identities follow from the chain rule of KL divergence. Since we have assumed that $\varepsilon$ follows a normal distribution with variance $\sigma^2$, the following equations hold:

$$\mathrm{KL}(\mu_{g_{(\mathbf{w}_{-j},0)}}(\cdot \mid p_{t-1}, x_{t-1}), \mu_{g_{(\mathbf{w}_{-j},1)}}(\cdot \mid p_{t-1}, x_{t-1}))$$
$$= \frac{1}{2\sigma^2}\left(\underline{b}p_{t-1} + g_{(\mathbf{w}_{-j},0)}(x_{t-1}) - \underline{b}p_{t-1} - g_{(\mathbf{w}_{-j},1)}(x_{t-1})\right)^2$$
$$= \frac{1}{2\sigma^2}\left(g_{(\mathbf{w}_{-j},0)}(x_{t-1}) - g_{(\mathbf{w}_{-j},1)}(x_{t-1})\right)^2$$
$$\leq \frac{1}{2\sigma^2}\left(\frac{1}{4M^{\beta}}\right)^2 \mathbb{I}_{\{x_{t-1} \in \mathbf{M}_j\}},$$

where the last inequality holds because $g_{(\mathbf{w}_{-j},0)}$ and $g_{(\mathbf{w}_{-j},1)}$ only differ in $\mathbf{M}_j$. Plugging the above equation into Eq. (26), we obtain

$$\mathrm{KL}(\mathbb{P}^{\pi,t}_{g_{(\mathbf{w}_{-j},0)}}, \mathbb{P}^{\pi,t}_{g_{(\mathbf{w}_{-j},1)}}) = \mathrm{KL}(\mathbb{P}^{\pi,t-1}_{g_{(\mathbf{w}_{-j},0)}}, \mathbb{P}^{\pi,t-1}_{g_{(\mathbf{w}_{-j},1)}}) + \frac{1}{2\sigma^2 M^d}\left(\frac{1}{4M^{\beta}}\right)^2$$
$$= \frac{1}{32\sigma^2 M^{d+2\beta}}t. \qquad (27)$$

where the first identity is holds since $\mathbb{E}_{\mathcal{P}_u \times \pi_{\mathcal{F}_{t-2}}, x_{t-1}}[\mathbb{I}_{\{x_{t-1} \in \mathbf{M}_j\}}] = \frac{1}{M^d}$, and the second identity follows by repeatedly applying the first identity. This completes the proof of Eq. (20).

Combining Eqs. (21), (22), (19), (24) and (27), we have

$$\sup_{g \in \mathcal{G}(\beta, d)} R_g^\pi(T) \geq \frac{|\underline{b}|}{2M^d M^d} \sum_{j=1}^{M^d} \sum_{\mathbf{w}_{-j} \in \{0,1\}^{M^d-1}} \sum_{t=1}^{T} \frac{1}{2} \left( \frac{1}{16M^\beta |\underline{b}|} \right)^2 \exp\left( -\frac{1}{32\sigma^2 M^{d+2\beta}} t \right)$$

$$\geq \frac{|\underline{b}|}{4} T \left( \frac{1}{16M^\beta |\underline{b}|} \right)^2 \exp\left( -\frac{1}{32\sigma^2 M^{d+2\beta}} T \right).$$

By letting $M = \lceil T^{\frac{1}{d+2\beta}} \rceil$ in the above inequality, we obtain the lower bound $\Omega(T^{\frac{d}{d+2\beta}})$.

### B.1. Proof of Lemma 1

We prove that for any $x, y \in [0,1]^d$, $|g_{\mathbf{w}}(x) - g_{\mathbf{w}}(y)| \leq 2||x - y||^\beta$ by considering two cases: $x$ and $y$ fall into the same bin in case 1, and $x$ and $y$ fall into different bins in case 2.

**Case 1**: $x, y \in \mathbf{M}_j$ for some $j \in [M^d]$. When $x$ and $y$ fall into the same bin $\mathbf{M}_j$, we divide the proof into four subcases.

Subcase 1.1: $w_j = 0$, or $w_j = 1$, $D(x, \partial \mathbf{M}_j) > \frac{1}{4^{1/\beta} M}$ and $D(y, \partial \mathbf{M}_j) > \frac{1}{4^{/\beta} M}$. In this subcase, we have $g_{\mathbf{w}}(x) = g_{\mathbf{w}}(y)$, and the result is trivial.

Subcase 1.2: $w_j = 1$, $D(x, \partial \mathbf{M}_j) \leq \frac{1}{4^{1/\beta} M}$ and $D(y, \partial \mathbf{M}_j) \leq \frac{1}{4^{1/\beta} M}$. Without loss of generality, we assume that $g_{\mathbf{w}}(x) \leq g_{\mathbf{w}}(y)$. Then we have the following equation:

$$||x - y||^\beta + g_{\mathbf{w}}(x) = ||x - y||^\beta + (D(x, \partial \mathbf{M}_j))^\beta + |\underline{b}|(\underline{p} + \overline{p})$$
$$= ||x - y||^\beta + \min_{z \in \partial \mathbf{M}_j} ||z - x||^\beta + |\underline{b}|(\underline{p} + \overline{p})$$
$$= \min_{z \in \partial \mathbf{M}_j} (||x - y||^\beta + ||z - x||^\beta) + |\underline{b}|(\underline{p} + \overline{p}). \tag{28}$$

If the following inequality holds: for $0 < \beta \leq 1$,

$$||a + b||^\beta \leq ||a||^\beta + ||b||^\beta, \forall a, b \in \mathbb{R}^d, \tag{29}$$

then we have from (28) that

$$||x - y||^\beta + g_{\mathbf{w}}(x) \geq \min_{z \in \partial \mathbf{M}_j} ||z - y||^\beta + |\underline{b}|(\underline{p} + \overline{p}) = (D(y, \partial \mathbf{M}_j))^\beta + |\underline{b}|(\underline{p} + \overline{p}) = g_{\mathbf{w}}(y),$$
$$\tag{30}$$

which then implies $|g_{\mathbf{w}}(x) - g_{\mathbf{w}}(y)| \leq ||x - y||^\beta$.

We now show (29). Note that (29) is simply the triangle inequality when $\beta = 1$. When $0 < \beta < 1$, let $\beta' \in \mathbb{R}^+$ be such that $\frac{1}{\beta} + \frac{1}{\beta'} = 1$. By applying Hölder's inequality, we have for any $a, b \in \mathbb{R}^+$,

$$(a + b)^\beta \leq \left( (a^\beta + b^\beta)^{\frac{1}{\beta}} (1^{\beta'} + 1^{\beta'})^{\frac{1}{\beta'}} \right)^\beta = 2^{\beta-1}(a^\beta + b^\beta) < a^\beta + b^\beta, \tag{31}$$

Applying (31) and the triangle inequality, we have for any $a, b \in \mathbb{R}^d$,

$$||a + b||^\beta \leq (||a|| + ||b||)^\beta < ||a||^\beta + ||b||^\beta,$$

which finishes the proof of (29).

Subcase 1.3: $w_j = 1$, $D(x, \partial \mathbf{M}_j) \leq \frac{1}{4^{1/\beta} M}$ and $D(y, \partial \mathbf{M}_j) > \frac{1}{4^{1/\beta} M}$. Let $\hat{\mathbf{M}}_j := \{z \in \mathbf{M}_j : D(z, \partial \mathbf{M}_j) > \frac{1}{4^{1/\beta} M}\}$. Since $\text{Proj}(x, \hat{\mathbf{M}}_j) \in \partial \hat{\mathbf{M}}_j$, then we have $g_{\mathbf{w}}(y) = g_{\mathbf{w}}(\text{Proj}(x, \hat{\mathbf{M}}_j)) = |\underline{b}|(\underline{p} + \overline{p}) + \frac{1}{4M^\beta}$ and

$$|g_{\mathbf{w}}(x) - g_{\mathbf{w}}(y)| = |g_{\mathbf{w}}(x) - g_{\mathbf{w}}(\text{Proj}(x, \hat{\mathbf{M}}_j))|. \tag{32}$$

Since $\partial\hat{\mathbf{M}}_j = \{z \in \mathbf{M}_j : D(z, \partial\mathbf{M}_j) = \frac{1}{4^{1/\beta}M}\}$ and $\mathrm{Proj}(x, \hat{\mathbf{M}}_j) \in \partial\hat{\mathbf{M}}_j$, it then follows that $D(\mathrm{Proj}(x, \hat{\mathbf{M}}_j), \partial\mathbf{M}_j) = \frac{1}{4^{1/\beta}M}$. Since $D(x, \partial\mathbf{M}_j) \leq \frac{1}{4^{1/\beta}M}$ from the assumption, by applying the result in subcase 1.2, we obtain

$$|g_{\mathbf{w}}(x) - g_{\mathbf{w}}(\mathrm{Proj}(x, \hat{\mathbf{M}}_j))| \leq ||x - \mathrm{Proj}(x, \hat{\mathbf{M}}_j)||^{\beta} = \min_{z \in \hat{\mathbf{M}}_j} ||x - z||^{\beta} \leq ||x - y||^{\beta}, \qquad (33)$$

where the last inequality holds due to $y \in \hat{\mathbf{M}}_j$. Combining (32) with (33), we obtain $|g_{\mathbf{w}}(x) - g_{\mathbf{w}}(y)| \leq ||x - y||^{\beta}$.

Subcase 1.4: $w_j = 1$, $D(x, \partial\mathbf{M}_j) > \frac{1}{4^{1/\beta}M}$ and $D(y, \partial\mathbf{M}_j) \leq \frac{1}{4^{1/\beta}M}$. The proof of this subcase is similar to subcase 1.3, and is omitted for brevity.

**Case 2**: $x \in \mathbf{M}_i$ and $y \in \mathbf{M}_j$ for $i \neq j$. When $x$ and $y$ fall into different bins, we divide the proof into five subcases.

Subcase 2.1: $w_i = w_j = 0$, or if $w_i = w_j = 1$, $D(x, \partial\mathbf{M}_i) > \frac{1}{4^{1/\beta}M}$ and $D(y, \partial\mathbf{M}_j) > \frac{1}{4^{1/\beta}M}$. In this subcase, we have $g_{\mathbf{w}}(x) = g_{\mathbf{w}}(y)$ and the result is trivial.

Subcase 2.2: $w_i = 0$ and $w_j = 1$. In this subcase, we have

$$|g_{\mathbf{w}}(x) - g_{\mathbf{w}}(y)| \leq (D(y, \partial\mathbf{M}_j))^{\beta} \qquad (34)$$

$$\leq ||\mathrm{Proj}(x, \partial\mathbf{M}_j) - y||^{\beta} \qquad (35)$$

$$\leq ||\mathrm{Proj}(x, \partial\mathbf{M}_j) - x||^{\beta} + ||x - y||^{\beta} \qquad (36)$$

$$= ||\mathrm{Proj}(x, \mathbf{M}_j) - x||^{\beta} + ||x - y||^{\beta} \qquad (37)$$

$$\leq 2||x - y||^{\beta}. \qquad (38)$$

In the above equations, Eq. (34) holds since under the assumption $w_i = 0$ and $w_j = 1$, if $D(y, \partial\mathbf{M}_j) \leq \frac{1}{4^{1/\beta}M}$, $|g_{\mathbf{w}}(x) - g_{\mathbf{w}}(y)| = (D(y, \partial\mathbf{M}_j))^{\beta}$, and if $D(y, \partial\mathbf{M}_j) > \frac{1}{4^{1/\beta}M}$, $|g_{\mathbf{w}}(x) - g_{\mathbf{w}}(y)| = \frac{1}{4M^{\beta}} \leq (D(y, \partial\mathbf{M}_j))^{\beta}$. Eq. (35) follows from the definition of $D(y, \partial\mathbf{M}_j)$ and $\mathrm{Proj}(x, \partial\mathbf{M}_j) \in \partial\mathbf{M}_j$. Eq. (36) follows from (29). Eq. (37) holds since if $\mathrm{Proj}(x, \mathbf{M}_j)$ is an interior point of $\mathbf{M}_j$, since $\mathbf{M}_j$ is a cubic, one can always construct a ball inside $\mathbf{M}_j$ with the center $\mathrm{Proj}(x, \mathbf{M}_j)$, and the intersected point between the ball and the line connecting $x$ and $\mathrm{Proj}(x, \mathbf{M}_j)$ has a strictly shorter distance to $x$ than $\mathrm{Proj}(x, \mathbf{M}_j)$, leading to contradiction with the fact that $\mathrm{Proj}(x, \mathbf{M}_j)$ is the closest point in the bin $\mathbf{M}_j$ to $x$. Thus, $\mathrm{Proj}(x, \mathbf{M}_j)$ must be at the boundary $\partial\mathbf{M}_j$ and $\mathrm{Proj}(x, \mathbf{M}_j) = \mathrm{Proj}(x, \partial\mathbf{M}_j)$. Eq. (38) follows from $y \in \mathbf{M}_j$ and the definition of $D(x, \partial\mathbf{M}_j)$.

Subcase 2.3: $w_i = 1$ and $w_j = 0$. The proof of this subcase is similar to subcase 2.2 and is omitted.

Subcase 2.4: $w_i = w_j = 1$, $D(x, \partial\mathbf{M}_i) \leq \frac{1}{4^{1/\beta}M}$ and $D(y, \partial\mathbf{M}_j) \leq \frac{1}{4^{1/\beta}M}$. Without loss of generality, we assume $g_{\mathbf{w}}(y) \geq g_{\mathbf{w}}(x)$. Then we have

$$||x - y||^{\beta} + g_{\mathbf{w}}(x) = \min_{z \in \partial\mathbf{M}_i} (||x - y||^{\beta} + ||z - x||^{\beta}) + |\underline{b}|(\underline{p} + \overline{p}) \geq \min_{z \in \partial\mathbf{M}_i} ||z - y||^{\beta} + |\underline{b}|(\underline{p} + \overline{p}), \qquad (39)$$

where the inequality follows from (29).

On the other hand, when $K$ is sufficiently large, we have

$$g_{\mathbf{w}}(y) = \min_{z \in \partial\mathbf{M}_j} ||z - y||^{\beta} + |\underline{b}|(\underline{p} + \overline{p})$$

$$= \min_{z \in \partial([-K,K]^d \setminus \mathrm{int}(\mathbf{M}_j))} ||z - y||^{\beta} + |\underline{b}|(\underline{p} + \overline{p}) \qquad (40)$$

$$= \min_{z \in [-K,K]^d \setminus \mathrm{int}(\mathbf{M}_j)} ||z - y||^{\beta} + |\underline{b}|(\underline{p} + \overline{p}) \qquad (41)$$

$$\leq \min_{z \in \partial\mathbf{M}_i} ||z - y||^{\beta} + |\underline{b}|(\underline{p} + \overline{p}). \qquad (42)$$

In the above equations, Eq. (40) holds since $\partial([-K, K]^d \setminus \mathrm{int}(\mathbf{M}_j)) = \partial([-K, K]^d) \cup \partial\mathbf{M}_j$, and when $K$ is sufficiently large, $D(y, \partial\mathbf{M}_j) < D(y, \partial([-K, K]^d))$ and $\mathrm{Proj}(y, \partial(\mathbf{M}_j)) =$

$\text{Proj}(y, \partial([-K, K]^d) \cup \partial \mathbf{M}_j)$. Eq. (41) holds due to the same reason as (37). Eq. (42) holds since $\partial(\mathbf{M}_i) \subset [-K, K]^d$ and $\partial(\mathbf{M}_i) \cap \text{int}(\mathbf{M}_j) = \emptyset$ imply that $\partial(\mathbf{M}_i) \subset [-K, K]^d \setminus \text{int}(\mathbf{M}_j)$.

Combining Eqs. (39) and (42), we obtain $|g_{\mathbf{w}}(x) - g_{\mathbf{w}}(x)| \leq ||x - y||^\beta$.

Subcase 2.5: $w_i = w_j = 1$, $D(x, \partial \mathbf{M}_j) \leq \frac{1}{4^{1/\beta} M}$ and $D(y, \partial \mathbf{M}_j) > \frac{1}{4^{1/\beta} M}$. In this subcase, we have

$$|g_{\mathbf{w}}(x) - g_{\mathbf{w}}(y)| = \left| (D(x, \partial \mathbf{M}_j))^\beta - \frac{1}{4M^\beta} \right| = \frac{1}{4M^\beta} - (D(x, \partial \mathbf{M}_j))^\beta < (D(y, \partial \mathbf{M}_j))^\beta - (D(x, \partial \mathbf{M}_j))^\beta.$$

The remaining analysis is similar to subcase 2.4, and is omitted for brevity.

## Appendix C. Proof of Theorem 3

We first establish the following lemma showing that the true demand function $f(p) + a^\top x$ can be well-approximated by the linear function $\hat{\theta}_{t,j}^\top \varphi(p) + \hat{a}_{t,j}^\top x$ within price segment $\mathbf{I}_j$ after running Algorithm 3. This result is quite standard and can be obtained easily by modifying the analysis in [1]. For completeness, we provide the details in Appendix C.1.

**Lemma 2** *For each $j \in [N]$, with probability at least $1 - \epsilon$, the following event holds: for any $t \in [T]$, $p \in \mathbf{I}$ and $x \in [0, 1]^d$,*

$$\left| f(p) + a^\top x - \left( \hat{\theta}_{t,j}^\top \varphi(p) + \hat{a}_{t,j}^\top x \right) \right| \leq \gamma_{t,j} \sqrt{\phi(p, x)^\top V_{t,j}^{-1} \phi(p, x)} + \Delta. \tag{43}$$

Now we highlight our key idea for analyzing the regret in each period $t \in [T]$. Let $p_t^* := \arg\max_{p \in [\underline{p}, \overline{p}]} p(f(p) + a^\top x_t)$ be the optimal price for period $t$ and $i_t^* \in [N]$ denote the index for the price segment $p_t^*$ belongs to. Conditioning on the events guaranteed by Lemma 2 for each price segment $i \in [N]$, we have

$$r_t = p_t^* \left( f(p_t^*) + a^\top x_t \right) - p_t \left( f(p_t) + a^\top x_t \right)$$

$$\leq \max_{p \in \mathbf{I}_{i_t^*}} p \left( \langle \hat{\theta}_{t,i_t^*}, \varphi(p) \rangle + \langle \hat{a}_{t,i_t^*}, x_t \rangle + \gamma_{t,i_t^*} \sqrt{\phi(p, x_t)^\top V_{t,i_t^*}^{-1} \phi(p, x_t)} + \Delta \right) - p_t \left( f(p_t) + a^\top x_t \right)$$

$$\leq \max_{i \in [N]} \max_{p \in \mathbf{I}_i} p \left( \langle \hat{\theta}_{t,i}, \varphi(p) \rangle + \langle \hat{a}_{t,i}, x_t \rangle + \gamma_{t,i} \sqrt{\phi(p, x_t)^\top V_{t,i}^{-1} \phi(p, x_t)} + \Delta \right) - p_t \left( f(p_t) + a^\top x_t \right)$$

$$= p_t \left( \langle \hat{\theta}_{t,i_t}, \varphi(p_t) \rangle + \langle \hat{a}_{t,i_t}, x_t \rangle + \gamma_{t,i_t} \sqrt{\phi(p_t, x_t)^\top V_{t,i_t}^{-1} \phi(p_t, x_t)} + \Delta \right) - p_t \left( f(p_t) + a^\top x_t \right)$$

$$\leq p_t \left| \langle \hat{\theta}_{t,i_t}, \varphi(p_t) \rangle + \langle \hat{a}_{t,i_t}, x_t \rangle - \left( f(p_t) + a^\top x_t \right) \right| + p_t \left( \gamma_{t,i_t} \sqrt{\phi(p_t, x_t)^\top V_{t,i_t}^{-1} \phi(p_t, x_t)} + \Delta \right)$$

$$\leq 2\overline{p} \left( \gamma_{t,i_t} \sqrt{\phi(p_t, x_t)^\top V_{t,i_t}^{-1} \phi(p_t, x_t)} + \Delta \right), \tag{44}$$

where the first inequality follows from Eq. (43) in Lemma 2 and $p_t^* \in \mathbf{I}_t^*$ by definition, the second equality is based on the design of our Algorithm 2 (line 13), and the last inequality again follows from Eq. (43) in Lemma 2. Therefore, by optimizing the revenue within each price segment in an optimistic way and choosing the price from all optimistic prices with the highest optimistic revenue, we reduce estimating the regret in period $t$ to bounding $\gamma_{t,i_t} \sqrt{\phi(p_t, x_t)^\top V_{t,i_t}^{-1} \phi(p_t, x_t)} + \Delta$. The subsequent analysis requires applying the elliptical potential lemma (see, e.g., [1]) to bound $\sqrt{\phi(p_t, x_t)^\top V_{t,i_t}^{-1} \phi(p_t, x_t)}$ and carefully choosing the number of price segments $N$ to balance the bias of polynomial approximation for $f(\cdot)$ and learning efficiency within each price segment. The details are given in Appendix B.1.

Let $n_{T,j} := \sum_{t=1}^T \mathbb{I}_{p_t \in \mathbf{I}_j}$ be the number of times for which prices $p_1, p_2, \ldots, p_T$ selected by our algorithm fall into $\mathbf{I}_j$, and $\mathcal{A}$ denote the event that Eq. (44) in Lemma 2 holds for any $t \in [T]$ and

$i \in [N]$. When $\mathcal{A}$ holds, from Eq. (44), the total regret can be bounded as follows:

$$\sum_{t=1}^{T} r_t \leq 2\overline{p} \sum_{t=1}^{T} \gamma_{t,i_t} \|\phi(p_t, x_t)\|_{V_{t,i_t}^{-1}} + 2\overline{p}\Delta T$$

$$= 2\overline{p} \sum_{j=1}^{N} \sum_{t=1}^{T} \gamma_{t,j} \|\phi(p_t, x_t)\|_{V_{t,j}^{-1}} \mathbb{I}_{p_t \in \mathbf{I}_j} + 2\overline{p}\Delta T$$

$$\leq 2\overline{p} \sum_{j=1}^{N} \gamma_{T,j} \sum_{t=1}^{T} \|\phi(p_t, x_t)\|_{V_{t,j}^{-1}} \mathbb{I}_{p_t \in \mathbf{I}_j} + 2\overline{p}\Delta T$$

$$\leq 2\overline{p} \sum_{j=1}^{N} \gamma_{T,j} \sqrt{n_{T,j}} \sqrt{\sum_{t=1}^{T} \|\phi(p_t, x_t)\|_{V_{t,j}^{-1}}^2 \mathbb{I}_{p_t \in \mathbf{I}_j}} + 2\overline{p}\Delta T$$

$$\leq 2\overline{p} \sum_{j=1}^{N} \gamma_{T,j} \sqrt{n_{T,j}} \sqrt{2(\mathfrak{b}(k) + d + 1) \ln\left(1 + \frac{n_{T,j}}{\mathfrak{b}(k) + d + 1}\right)} + 2\overline{p}\Delta T, \qquad (45)$$

where the second inequality holds since $\{\gamma_{t,j} : 1 \leq t \leq T\}$ is an increasing sequence for each $j \in [N]$, the third inequality follows from Cauchy-Schwarz inequality, and the last inequality holds due to $\sum_{t=1}^{T} \|\phi(p_t, x_t)\|_{V_{t,j}^{-1}}^2 \mathbb{I}_{p_t \in \mathbf{I}_j} \leq 2 \ln \det(V_{T,j})/\ln(\lambda I)$ from the elliptical potential lemma (see, e.g., Lemma 11 in [1]) and $\ln \det(V_{T,j}) \leq (\mathfrak{b}(k) + d + 1) \ln(\lambda(1 + \frac{n_{T,j}}{\mathfrak{b}(k) + d + 1}))$ from (51).

Since $\epsilon = T^{-2}$, we then have

$$\gamma_{T,j} = \sigma \sqrt{(\mathfrak{b}(k) + d + 1) \ln\left(\frac{\mathfrak{b}(k) + d + 1 + n_{T,j}}{\mathfrak{b}(k) + d + 1}\right) - 2\ln \epsilon} + \lambda^{\frac{1}{2}} (C_0^2(\mathfrak{b}(k) + 1) + \overline{a}^2)^{\frac{1}{2}} + \Delta\sqrt{n_{T,j}}$$

$$\leq \sigma \sqrt{2(\mathfrak{b}(k) + d + 1)\ln(T+1)} + \lambda^{\frac{1}{2}} (C_0^2(\mathfrak{b}(k) + 1) + \overline{a}^2)^{\frac{1}{2}} + \Delta\sqrt{n_{T,j}},$$

and the first term in the RHS of (45) is bounded by

$$\sum_{j=1}^{N} \gamma_{T,j} \sqrt{n_{T,j}} \sqrt{2(\mathfrak{b}(k) + d + 1) \ln\left(1 + \frac{n_{T,j}}{\mathfrak{b}(k) + d + 1}\right)}$$

$$\leq 2\sqrt{(\mathfrak{b}(k) + d + 1)\ln(T+1)} \left(\max\left\{\sigma\sqrt{(\mathfrak{b}(k) + d + 1)\ln(T+1)}, \lambda^{\frac{1}{2}}(C_0^2 k + \overline{a}^2)^{\frac{1}{2}}\right\} \cdot \sum_{j=1}^{N} \sqrt{n_{T,j}} + \Delta T\right)$$

$$\leq 2\sqrt{(\mathfrak{b}(k) + d + 1)\ln(T+1)}$$
$$\times \left(\max\left\{\sigma\sqrt{(\mathfrak{b}(k) + d + 1)\ln(T+1)}, \lambda^{\frac{1}{2}}(C_0^2(\mathfrak{b}(k) + 1) + \overline{a}^2)^{\frac{1}{2}}\right\} \cdot \sqrt{\sum_{j=1}^{N} n_{T,j}} \cdot \sqrt{\sum_{j=1}^{N} 1^2} + \Delta T\right)$$

$$= 2\sqrt{(\mathfrak{b}(k) + d + 1)\ln(T+1)} \left(\max\left\{\sigma\sqrt{(\mathfrak{b}(k) + d + 1)\ln(T+1)}, \lambda^{\frac{1}{2}}(C_0^2(\mathfrak{b}(k) + 1) + \overline{a}^2)^{\frac{1}{2}}\right\} \cdot \sqrt{TN}$$
$$+ \Delta T\right).$$

This, together with $\Delta = \Theta(\frac{\delta}{N^k})$ and (45), implies

$$\sum_{t=1}^{T} r_t = \mathcal{O}\left(\sqrt{(\mathfrak{b}(k) + d + 1)\ln T}\right) \cdot \mathcal{O}\left(\sqrt{(\mathfrak{b}(k) + d + 1)NT \ln T} + \frac{\delta T}{N^k}\right).$$

To balance the two terms $\sqrt{(\mathfrak{b}(k) + d + 1)NT \ln T}$ and $\frac{\delta T}{N^k}$, we let $N = \lceil (T\delta^2)^{\frac{1}{2k+1}} \rceil + 1$. When $\delta = \mathcal{O}(T^{-\frac{1}{2}})$, $N = \Theta(1)$ and thus we obtain $\sqrt{(\mathfrak{b}(k) + d + 1)NT} = \Theta(\sqrt{(\mathfrak{b}(k) + d + 1)T})$ and $\frac{\delta T}{N^k} = \mathcal{O}(\sqrt{T})$. In this case, we get $\sum_{t=1}^{T} r_t = \mathcal{O}((\mathfrak{b}(k) + d + 1)\sqrt{T} \ln T)$. When $\delta = \Omega(T^{-\frac{1}{2}})$, $N = \Theta((T\delta^2)^{\frac{1}{2k+1}})$ and we obtain $\sum_{t=1}^{T} r_t = \mathcal{O}((\mathfrak{b}(k) + d + 1)\delta^{\frac{1}{2k+1}} T^{\frac{k+1}{2k+1}} \ln T)$. Combining these two cases, we have $\sum_{t=1}^{T} r_t = \mathcal{O}((\mathfrak{b}(k) + d + 1)((\delta T^{k+1})^{\frac{1}{2k+1}} \vee \sqrt{T}) \ln T)$.

Therefore, the total expected regret is upper bounded by

$$\sum_{t=1}^{T} \mathbb{E}[r_t] = \sum_{t=1}^{T} \mathbb{E}[r_t|\mathcal{A}] \cdot \mathbb{P}(\mathcal{A}) + \sum_{t=1}^{T} \mathbb{E}[r_t|\mathcal{A}^c] \cdot \mathbb{P}(\mathcal{A}^c)$$

$$= \widetilde{\mathcal{O}}\left( (\mathfrak{b}(k) + d + 1) \left( (\delta T^{k+1})^{\frac{1}{2k+1}} \vee \sqrt{T} \right) \right) + \mathcal{O}\left( \frac{N}{T} \right), \qquad (46)$$

where the second identity holds since from the union bound, $\mathbb{P}(\mathcal{A}^c) \leq \frac{N}{\epsilon} = \frac{N}{T^2}$. Since $\mathcal{O}(\frac{N}{T}) = \mathcal{O}(T^{-\frac{2k}{2k+1}} \delta^{\frac{2}{2k+1}})$ and $\delta = \mathcal{O}(1)$, the first term in the RHS of (46) dominates.

## C.1. Proof of Lemma 2

In this proof, we consider an arbitrarily fixed $\mathbf{I} \in \bigcup_{j \in [N]} \mathbf{I}_j$. Let $P_{\mathbf{I}}(p)$ be the first $\mathfrak{b}(k) + 1$ terms of the Taylor expansion for function $f(p)$ at $p = l$:

$$P_{\mathbf{I}}(p) = \sum_{i=0}^{\mathfrak{b}(k)} \frac{f^{(i)}(l)}{i!} (p - l)^i, \quad \forall p \in \mathbf{I}.$$

It's easy to verify that $|f(p) - P_{\mathbf{I}}(p)| \leq \delta \frac{(u-l)^k}{\mathfrak{b}(k)!}$. For convenience, we label $(x, p, d)$ in $\mathcal{D}$ as $\{(x_i, p_i, d_i)\}_{i=1}^{t}$ in chronological order. Let $\beta_i := f(p_i) - P_{\mathbf{I}}(p_i)$ for $i \in [t]$, then $|\beta_i| \leq \Delta$ and $d_i = f(p_i) + a^\top x_i + \epsilon_i = P_I(p_i) + a^\top x_i + \epsilon_i + \beta_i$.

Denote $\boldsymbol{d} = (d_i)_{i \leq t}, \boldsymbol{\epsilon} = (\epsilon_i)_{i \leq t}, \boldsymbol{\beta} = (\beta_i)_{i \leq t}$ as column vectors in $\mathbb{R}^t$. Let $\boldsymbol{X} = ((\phi(p_i, x_i))_{i \leq t}) \in \mathbb{R}^{t \times (\mathfrak{b}(k)+d+1)}, \theta^* = ((\frac{f^{(i)}(l)}{i!})_{0 \leq i \leq \mathfrak{b}(k)}, a) \in \mathbb{R}^{\mathfrak{b}(k)+d+1}$. Thus, $\boldsymbol{d} = \boldsymbol{X}\theta^* + \boldsymbol{\epsilon} + \boldsymbol{\beta}$. The Ridge estimate $\bar{\theta} := (\hat{\theta}^\top, \hat{a}^\top)^\top$ can be written as $\bar{\theta} = V^{-1}\boldsymbol{X}^\top \boldsymbol{d}$. Note that we omit the dependency of $\bar{\theta}$, $V$, $\boldsymbol{X}$ and $\boldsymbol{d}$ on the cardinality of $\mathcal{D}$, i.e., $t$, for simplicity. By simple calculation, we have

$$\bar{\theta} - \theta^* = -\lambda V^{-1}\theta^* + V^{-1}\boldsymbol{X}^\top(\boldsymbol{\epsilon} + \boldsymbol{\beta}).$$

Multiplying $(\bar{\theta} - \theta^*)^\top V$ on both sides of the above equation, we have

$$(\bar{\theta} - \theta^*)^\top V (\bar{\theta} - \theta^*)$$
$$= -\lambda(\bar{\theta} - \theta^*)^\top \theta^* + (\bar{\theta} - \theta^*)^\top \boldsymbol{X}^\top (\boldsymbol{\epsilon} + \boldsymbol{\beta})$$
$$\leq \lambda \left\| (\bar{\theta} - \theta^*) \right\|_V \cdot \|\theta^*\|_{V^{-1}} + \left\| (\bar{\theta} - \theta^*) \right\|_V \cdot \left\| \boldsymbol{X}^\top \boldsymbol{\epsilon} \right\|_{V^{-1}} + (\bar{\theta} - \theta^*)^\top \boldsymbol{X}^\top \boldsymbol{\beta}$$
$$\leq \lambda \left\| (\bar{\theta} - \theta^*) \right\|_V \cdot \|\theta^*\|_{V^{-1}} + \left\| (\bar{\theta} - \theta^*) \right\|_V \cdot \left\| \boldsymbol{X}^\top \boldsymbol{\epsilon} \right\|_{V^{-1}} + \Delta\sqrt{t} \left\| (\bar{\theta} - \theta^*) \right\|_V, \qquad (47)$$

where the first inequality follows from Cauchy-Schwarz inequality, and the second inequality holds due to

$$\left| (\bar{\theta} - \theta^*)^\top \boldsymbol{X}^\top \boldsymbol{\beta} \right| \leq \sqrt{\sum_{1 \leq \tau \leq t} \beta_\tau^2} \sqrt{\sum_{1 \leq \tau \leq t} \left( \langle \phi(p_\tau, x_\tau), \bar{\theta} - \theta^* \rangle \right)^2}$$

$$= \sqrt{\sum_{1 \leq \tau \leq t} \beta_\tau^2} \sqrt{(\bar{\theta} - \theta^*)^\top \left( \sum_{1 \leq \tau \leq t} \phi(p_\tau, x_\tau)\phi^\top(p_\tau, x_\tau) \right) (\bar{\theta} - \theta^*)}$$

$$= \sqrt{\sum_{1 \leq \tau \leq t} \beta_\tau^2} \sqrt{(\bar{\theta} - \theta^*)^\top (V - \lambda I) (\bar{\theta} - \theta^*)}$$

$$\leq \Delta\sqrt{t} \left\| (\bar{\theta} - \theta^*) \right\|_V.$$

Dividing both sides of Eq. (47) by $\left\| \bar{\theta} - \theta^* \right\|_V$, we have

$$\left\| (\bar{\theta} - \theta^*) \right\|_V \leq \lambda \|\theta^*\|_{V^{-1}} + \left\| \boldsymbol{X}^\top \boldsymbol{\epsilon} \right\|_{V^{-1}} + \Delta\sqrt{t}. \qquad (48)$$

By applying the self-normalized bound for vector-valued martingales in Theorem 1 of [1], we have with probability at least $1 - \epsilon$,

$$\forall t \in [T]: \left\| \boldsymbol{X}^\top \boldsymbol{\epsilon} \right\|_{V^{-1}} \leq \sqrt{2\sigma^2 \log \left( \frac{\det(V)^{\frac{1}{2}} \det(\lambda I)^{-1/2}}{\epsilon} \right)}. \qquad (49)$$

Since $\|\theta^*\|_{V^{-1}}^2 \leq \frac{1}{\lambda_{\min}(V)}\|\theta^*\|_2^2 \leq \frac{1}{\lambda}\|\theta^*\|_2^2 \leq \frac{C_0^2(\mathfrak{b}(k)+1)+\bar{a}^2}{\lambda}$, it follows from Eqs. (48) and (49) that

$$\left\|(\bar{\theta}-\theta^*)\right\|_V \leq \sqrt{2\sigma^2 \log\left(\frac{\det(V)^{\frac{1}{2}}\det(\lambda I)^{-1/2}}{\delta}\right)} + \lambda^{\frac{1}{2}}(C_0^2(\mathfrak{b}(k)+1)+\bar{a}^2)^{\frac{1}{2}} + \Delta\sqrt{t}. \quad (50)$$

Moreover, let $\lambda_1, \lambda_2, \ldots, \lambda_{\mathfrak{b}(k)+d+1}$ denote the eigenvalues of matrix $V$. Then we have

$$\begin{aligned}
\det V &= \Pi_{i=1}^{\mathfrak{b}(k)+d+1}\lambda_i \\
&\leq \left(\frac{\sum_{i=1}^{\mathfrak{b}(k)+d+1}\lambda_i}{\mathfrak{b}(k)+d+1}\right)^{\mathfrak{b}(k)+d+1} \\
&= \left(\lambda + \frac{\sum_{i=1}^{t}\|\phi(p_i,x_i)\|_2^2}{\mathfrak{b}(k)+d+1}\right)^{\mathfrak{b}(k)+d+1} \\
&\leq \left(\lambda + \frac{|\mathcal{D}|\lambda}{\mathfrak{b}(k)+d+1}\right)^{\mathfrak{b}(k)+d+1}, \quad (51)
\end{aligned}$$

where the first equality follows from the fact that the determinant of a matrix equals the product of its all eigenvalues, the second inequality follows from the inequality of arithmetic and geometric means, the second equality follows from the fact that the sum of eigenvalues of a matrix equals its trace, and the last inequality holds since $\|\phi(p_i,x_i)\|^2 = \sum_{j=0}^{\mathfrak{b}(k)}(p_i-l)^j + \|x_i\|^2 \leq \frac{1-(u-l)^{2\mathfrak{b}(k)}}{1-(u-l)^2} + d \leq \frac{1}{1-(\bar{p}-\underline{p})^2/N^2} + d = \lambda$ and $t = |\mathcal{D}|$.

Therefore, with probability at least $1 - \epsilon$, for any $t \in [T]$, $p \in \mathbf{I}$ and $x \in [0,1]^d$,

$$\begin{aligned}
\left|f(p) + a^\top x - (\hat{\theta}^\top\varphi(p) + \hat{a}^\top x)\right| &\leq \left|P_I(p) + a^\top x - (\hat{\theta}^\top\varphi(p) + \hat{a}^\top x)\right| + |f(p) - P_I(p)| \\
&\leq \left\|(\bar{\theta}-\theta^*)\right\|_V\|\phi(p,x)\|_{V^{-1}} + \Delta \\
&\leq \gamma\|\phi(p,x)\|_{V^{-1}} + \Delta,
\end{aligned}$$

where the first inequality follows from the triangle inequality, the second inequality follows from Cauchy-Schwarz inequality, and the last inequality follows from Eqs. (50) and (51) and the definition of $\gamma$.

## Appendix D. Proof of Theorem 4

The worst-case bound $\Omega(\sqrt{T})$ is directly implied from Theorem 1 in [18] by letting $a = \mathbf{0}$ and $f(p) = \alpha + \beta p$ (which belongs to $\mathcal{F}^k([\underline{p},\bar{p}];\delta)$) when $\alpha$ and $\beta$ are appropriately defined to adapt to $\delta$). To show the instance-dependent bound $\Omega((\delta T^{k+1})^{\frac{1}{2k+1}})$, we first construct a series of demand functions and use the Kullback-Leibler (KL) divergence arguments to bound the regret. Note that the smoothstep function adopted by [32] can not be used here because in our problem $k$ is not necessarily a constant. Besides, our analysis also differs from [32] in that the constructed demand functions need to be instance-dependent such that the established lower bound achieves a tight dependency on $\delta$.

Specifically, following from a similar idea of [15], we start with introducing an infinitely differentiable function $u(x)$ defined as

$$u(x) := \begin{cases} \exp\left\{-\frac{1}{x(1-x)}\right\} & \text{if } x \in [0,1]; \\ 0 & \text{otherwise}. \end{cases}$$

Consider $S : \mathbb{R}^+ \to \mathbb{R}^+$ as follows:

$$S(x) = \left(\int_0^2 u(t)dt\right)^{-1}\int_{-\infty}^x u(t)dt. \quad (52)$$

Note that $S(x)$ is non-decreasing infinitely differentiable function satisfying $S(x) = 0$ on $(-\infty, 0]$ and $S(x) = 1$ on $[1, \infty)$. For any integer $l \geq 1$, the $l$-th derivative of $S(x)$ at $x \in [0,1]$ is in the form

of $\frac{poly(x)}{(x(1-x))^{2(l-1)}}\exp\left(-\frac{1}{x(1-x)}\right)$, which is bounded in the domain. Moreover, $S^{(l)}(1) = S^{(l)}(0) = 0$. Besides, there exist a constant $c_k$, such that for any $x, x' \in \mathbb{R}$, $|S^{(\mathfrak{b}(k))}(x) - S^{(\mathfrak{b}(k))}(x')| \leq c_k|x - x'|^{k-\mathfrak{b}(k)}$. Based on $S(x)$, we can define $g(\cdot) : [0, 2] \to [0, \frac{1}{Z_k}]$ as follows:

$$g_k(x) = \frac{1}{Z_k}(S(x)\mathbf{1}\{x \leq 1\} + S(2-x)\mathbf{1}\{x > 1\}), \tag{53}$$

where $Z_k > 0$ is a scaling parameter which makes the $l$-th derivatives of $g_k(x)$ uniformly bounded by constant 1 on $[0, 2]$ for each $0 \leq l \leq \mathfrak{b}(k)$, and $|g_k^{(\mathfrak{b}(k))}(x) - g_k^{(\mathfrak{b}(k))}(x')| \leq |x - x'|^{k-\mathfrak{b}(k)}$ for any $x, x' \in \mathbb{R}$, e.g., $Z_k := \max_{0 \leq k' \leq k}\max_{x \in [0,1]}|S^{(k)}(x)| \vee c_k$. Since $S(1) = 1$, $Z_k \geq 1$ naturally holds.

We partition the price range $[\underline{p}, \overline{p}]$ into $J$ segments of equal length, denoted by $\mathbf{I}_1, \mathbf{I}_2, \ldots, \mathbf{I}_J$. We construct a series of revenue functions $r_0, r_1, r_2, \ldots, r_J$ as following,

$$r_j(p) := \begin{cases} \frac{1}{2}\hat{\delta} & \text{if } p \notin \mathbf{I}_j; \\ \frac{1}{2}\hat{\delta} + \eta \cdot g_k\big(2J(p - a_j)\big) & \text{if } p \in \mathbf{I}_j, \end{cases} \tag{54}$$

where $\hat{\delta} = \delta/((\sum_{i=0}^{\mathfrak{b}(k)}\frac{\mathfrak{b}(k)!}{i!}) \vee (\mathfrak{b}(k)+1)2^{\mathfrak{b}(k)-1})$. By choosing $J := \lceil 4(\mathfrak{b}(k)+1)2^{\mathfrak{b}(k)}\hat{\delta}^{\frac{2}{2k+1}}T^{\frac{1}{2k+1}}\rceil$ and $\eta = ((2\sigma) \wedge \frac{1}{2^{3k+1}})\frac{1}{((\mathfrak{b}(k)+1)2^{\mathfrak{b}(k)})^k}\hat{\delta}^{\frac{1}{2k+1}}T^{-\frac{k}{2k+1}}$, we can have the following lemma on the demand function $f_j(p) = \frac{f_j(p)}{p}$, whose proof is provided in Appendix D.1.

**Lemma 3** *For each $0 \leq j \leq J$, $f_j(p) \in \mathcal{F}^k([\underline{p}, \overline{p}]; \delta)$.*

Note that for each $j \in [J]$, the induced optimal price of $f_j(p)$ belongs to $\mathbf{I}_j$ and $r_j(p)$ differs from $r_0(p)$ only in $\mathbf{I}_j$, with the maximum difference characterized by parameter $\eta$. It's important to note that $\eta$ is a crucial quantity that balances the tradeoff between making it a more challenging task for the algorithm to distinguish between different demand functions (which requires $\eta$ to be small) and imposing a higher regret loss if the algorithm fails to identify the true demand environment (which requires $\eta$ to be large). For any policy $\pi$, consider the random variable $T_j$ denoting the number of times the prices selected by $\pi$ fall into segment $\mathbf{I}_j$. Similar to [32], we establish the following inequalities: for each $j \in [J]$,

$$\left|\mathbb{E}_j^\pi[T_j] - \mathbb{E}_0^\pi[T_j]\right| \leq \frac{1}{2}T\sqrt{\mathrm{KL}(\mathbb{P}_0^\pi(T_j)||\mathbb{P}_j^\pi(T_j))} \leq \frac{1}{4\sigma}\sqrt{\mathbb{E}_0^\pi[T_j]}T\eta. \tag{55}$$

In Eq. (55), the first inequality is obtained by bounding $\left|\mathbb{E}_j^\pi[T_j] - \mathbb{E}_0^\pi[T_j]\right|$ via the total variation of $\mathbb{P}_0^\pi$ and $\mathbb{P}_j^\pi$ and applying Pinsker's inequality that relates the total variation of two probability measures with the KL divergence, and the second inequality is due to our construction of $r_j$. Letting $j^* := \arg\min_{1 \leq j \leq J}\mathbb{E}_0^\pi[T_j]$, we must have $\mathbb{E}_0^\pi[T_j] \leq \frac{T}{J}$. Since we set $\eta = \Theta((\delta T^{-k})^{\frac{1}{2k+1}})$ and $J = \Theta((\delta^2 T)^{\frac{1}{2k+1}})$, Eq. (55) guarantees that $\mathbb{E}_{j^*}^\pi[T_{j^*}] \leq \frac{T}{2}$. This indicates that when the true revenue function is $r_{j^*}$, there are at least $\frac{T}{2}$ times when the selected prices do not fall into the "best" segment $\mathbf{I}_{j^*}$, leading to the regret loss $\Omega(T\eta) = \Omega((\delta T^{k+1})^{\frac{1}{2k+1}})$.

As discussed before, it suffices to prove the lower bound $\Omega((\delta T^{k+1})^{\frac{1}{2k+1}})$. Note that we also only need to consider the case $\delta \geq T^{-\frac{1}{2}}$, since otherwise, $(\delta T^{k+1})^{\frac{1}{2k+1}} < \sqrt{T}$, and the desired lower bound in (2) becomes $\Omega(\sqrt{T})$, which is again obtained. For simplicity, we assume $[\underline{p}, \overline{p}] = [1, 2]$ and $\mathcal{D}$ is a standard normal distribution.

Based on what we have constructed, for any given policy $\pi$, let $T_j$ be the number of times when the prices selected by $\pi$ fall into segment $\mathbf{I}_j$. We then claim the following inequality for any $1 \leq j \leq J$:

$$\left|\mathbb{E}_0^\pi[T_j] - \mathbb{E}_j^\pi[T_j]\right| \leq \frac{1}{4\sigma}\sqrt{\mathbb{E}_0^\pi[T_j]}T\eta, \tag{56}$$

where $\mathbb{E}_0^\pi[\cdot]$ and $\mathbb{E}_j^\pi[\cdot]$ denote the expectation associated with the probability measure induced by policy $\pi$ under demand model $r_0$ and $r_j$ respectively. The proof of (56) is deferred to the last part,

and we now proceed to prove the regret lower bound based on (56). Consider the index $j^* \in [J]$ that minimizes $\mathbb{E}_0^\pi[T_{j^*}]$. By Pigeonhole principle, we have $\mathbb{E}_0^\pi[T_{j^*}] \leq T/J$. From (56), we further have

$$\mathbb{E}_{j^*}^\pi[T_{j^*}] \leq \frac{1}{4\sigma}\sqrt{\mathbb{E}_0^\pi[T_{j^*}]}T\eta + \mathbb{E}_0^\pi[T_{j^*}] \leq \frac{1}{4\sigma}\sqrt{\frac{T}{J}}T\eta + \frac{T}{J} \leq \frac{T}{2}, \qquad (57)$$

where the first inequality follows from (56), the second inequality follows from the choice of $j$, and the last inequality holds since $\eta^2 \leq (2\sigma\hat{\delta}^{\frac{1}{2k+1}}T^{-\frac{k}{2k+1}})^2 = 4\sigma^2\hat{\delta}^{\frac{2}{2k+1}}T^{-\frac{2k}{2k+1}} \leq \sigma^2\frac{J}{T}$ implies $\frac{1}{4\sigma}\sqrt{\frac{T}{J}}T\eta \leq \frac{1}{4}T$ and $\hat{\delta} \geq T^{-\frac{1}{2}}$ implies $\frac{T}{J} = \frac{1}{4}\hat{\delta}^{-\frac{2}{2k+1}}T^{\frac{2k}{2k+1}} \leq \frac{1}{4}T$. Note that when the true demand function is $f_{j^*}(\cdot)$, in any period when policy $\pi$ charges a price out of $\mathbf{I}_j$, a revenue loss $\eta$ will be incurred by the definition of $r_j(\cdot)$. Hence, we have

$$\sup_{f \in \{f_1, f_2, \ldots, f_J\}} R_{f_j}^\pi(T, k, \delta) \geq R_{f_{j^*}}^\pi(T, k, \delta) \geq (T - \mathbb{E}_{j^*}^\pi[T_{j^*}])\eta \geq \frac{1}{2}T\eta = \Omega\left(\left(\delta T^{k+1}\right)^{\frac{1}{2k+1}}\right).$$

Finally, we complete the proof of Theorem 4 by proving (56). For the sake of rigor, we define a probability space as follows. Let $\Omega = ([1, 2] \times \mathbb{R})^T \times \{0, 1, 2, \ldots, T\}$ and $\mathcal{B}(\Omega)$ be the Borel algebra on $\Omega$. For any $t \in [T]$, let $P_t$ and $D_t$ be measurable functions on $(\Omega, \mathcal{B}(\Omega))$ that map each $\omega = (p_1, d_1, p_2, d_2, \ldots, p_T, d_T) \in \Omega$ to $p_t$ and $d_t$ respectively. For any $j \in \{0\} \cup [J]$, let $T_j$ be a measurable function on $(\Omega, \mathcal{B}(\Omega))$ that maps $\omega = (p_1, d_1, p_2, d_2, \ldots, p_T, d_T) \in \Omega$ to the cardinality of the set $\{1 \leq t \leq T : p_t \in \mathbf{I}_j\}$. We also define two functions $\mu_i^\pi : ([1, 2] \times \mathbb{R})^T \to \mathbb{R}^+$ and $\nu_i^\pi : ([1, 2] \times \mathbb{R})^T \times \{0, 1, 2, \ldots, T\} \to \mathbb{R}^+$ as follows:

$$\nu_j^\pi(p_1, d_1, p_2, d_2, \ldots, p_T, d_T) = \prod_{t=1}^T \left(\mu^\pi(p_t|p_1, d_1, \ldots, p_{t-1}, d_{t-1}) \cdot \frac{1}{\sqrt{2\pi}\sigma}e^{-\frac{(d_t - f_j(p_t))^2}{2\sigma^2}}\right),$$

$$\mu_j^\pi(p_1, d_1, p_2, d_2, \ldots, p_T, d_T, t_j) = \nu_j^\pi(p_1, d_1, p_2, d_2, \ldots, p_T, d_T) \cdot \mathbf{1}_{\{t_j = |1 \leq t \leq T : p_t \in \mathbf{I}_j|\}},$$

where $\mu^\pi(p_t|p_1, d_1, \ldots, p_{t-1}, d_{t-1})$ is the p.d.f. for $p_t$ given $(p_1, d_1, \ldots, p_{t-1}, d_{t-1})$. Let $\mathbb{P}_j^\pi(\cdot)$ be the following probability measure on $(\Omega, \mathcal{B}(\Omega))$: for any $B \in \mathcal{B}(\Omega)$, $\mathbb{P}_j^\pi(B) = \int_B \mu_j^\pi(w)dw$. Thus, $(\Omega, \mathcal{B}(\Omega), \mathbb{P}_j^\pi)$ constitute a probability space, and from the chain rule, $\nu_j^\pi(\cdot)$ and $\mu_j^\pi(\cdot)$ are the p.d.f. for $(P_1, D_1, P_2, D_2, \ldots, P_T, D_T)$ and $(P_1, D_1, P_2, D_2, \ldots, P_T, D_T, T_j)$ respectively. With a slight abuse of notation, we denote the distributions of $T_j$ and $(P_1, D_1, P_2, D_2, \ldots, P_T, D_T)$ by $\mathbb{P}_i^\pi(T_j)$ and $\mathbb{P}_i^\pi(P_1, D_1, P_2, D_2, \ldots, P_T, D_T)$ respectively, and the conditional probability distribution of $T_j$ given $(P_1, D_1, P_2, D_2, \ldots, P_T, D_T)$ by $\mathbb{P}_i^\pi(T_j|P_1, D_1, P_2, D_2, \ldots, P_T, D_T)$. For any given $0 \leq i \leq J$ and $0 \leq j \leq J$, $\mathbb{E}_i^\pi[T_j]$ is then the expectation of $T_j$ under $\mathbb{P}_i^\pi$.

Then we note that

$$\left|\mathbb{E}_0^\pi[T_j] - \mathbb{E}_j^\pi[T_j]\right| \leq \sum_{t=0}^T t \times \left|\mathbb{P}_0^\pi(t) - \mathbb{P}_j^\pi(t)\right| \leq T \times \sum_{t=0}^T \left|\mathbb{P}_0^\pi(t) - \mathbb{P}_j^\pi(t)\right|$$

$$= \frac{1}{2}T\|\mathbb{P}_0^\pi(T_j) - \mathbb{P}_j^\pi(T_j)\|_{\text{TV}} \leq \frac{1}{2}T\sqrt{\frac{1}{2}\text{KL}(\mathbb{P}_0^\pi(T_j)\|\mathbb{P}_j^\pi(T_j))}, \qquad (58)$$

where the first identity follows from the property of the total variation distance for discrete random variables, see, e.g., Proposition 4.2 in [19], and the last inequality follows from Pinsker's inequality.

To further bound $\mathrm{KL}(\mathbb{P}_0^\pi(T_j)||\mathbb{P}_j^\pi(T_j))$, we note that

$$
\begin{aligned}
\mathrm{KL}\Big(\mathbb{P}_0^\pi(T_j)\,||\,\mathbb{P}_j^\pi(T_j)\Big) &= \mathrm{KL}\Big(\mathbb{P}_0^\pi(P_1,D_1,P_2,D_2,\ldots,P_T,D_T,T_j)\,||\,\mathbb{P}_j^\pi(P_1,D_1,P_2,D_2,\ldots,P_T,D_T,T_j)\Big) \\
&\quad - \mathrm{KL}\Big(\mathbb{P}_0^\pi(P_1,D_1,P_2,D_2,\ldots,P_T,D_T|T_j)\,||\,\mathbb{P}_j^\pi(P_1,D_1,P_2,D_2,\ldots,p_T,D_T|T_j)\Big) \\
&\leq \mathrm{KL}\Big(\mathbb{P}_0^\pi(P_1,D_1,P_2,D_2,\ldots,P_T,D_T,T_j)\,||\,\mathbb{P}_j^\pi(P_1,D_1,P_2,D_2,\ldots,P_T,D_T,T_j)\Big) \\
&= \mathbb{E}_0^\pi\left[\mathbb{E}_0^\pi\left[\log\frac{\mu_0^\pi(P_1,D_1,P_2,D_2,\ldots,P_T,D_T,T_j)}{\mu_j^\pi(P_1,D_1,P_2,D_2,\ldots,P_T,D_T,T_j)}\Big|(P_1,D_1,\ldots,P_T,D_T)\right]\right] \\
&= \mathbb{E}_0^\pi\left[\log\frac{\nu_0^\pi(P_1,D_1,P_2,D_2,\ldots,P_T,D_T)}{\nu_j^\pi(P_1,D_1,P_2,D_2,\ldots,P_T,D_T)}\right] \\
&= \frac{1}{2\sigma^2}\sum_{t=1}^T\mathbb{E}_0^\pi\left[(D_t-f_j(P_t))^2-(D_t-f_0(P_t))^2\right] \\
&= \frac{1}{2\sigma^2}\sum_{t=1}^T\mathbb{E}_0^\pi\left[(f_0(P_t)-f_j(P_t))^2\right] \\
&= \frac{1}{2\sigma^2}\sum_{t=1}^T\mathbb{E}_0^\pi\left[\mathbf{1}_{\{P_t\in\mathbf{I}_j\}}(f_0(P_t)-f_j(P_t))^2\right] \\
&\leq \frac{1}{2\sigma^2}\mathbb{E}_0^\pi[T_j]\cdot\max_{p\in\mathbf{I}_j}(f_0(p)-f_j(p))^2 \\
&= \frac{1}{2\sigma^2}\mathbb{E}_0^\pi[T_j]\eta^2,
\end{aligned}
\tag{59}
$$

where the first identity follows from the chain rule for KL divergence, the first inequality follows from the fact that the KL divergence between any two probability distributions is non-negative, the second identity holds due to the definition of KL divergence and the law of total expectation, the third identity holds since given $(P_1,D_1,P_2,D_2,\ldots,P_T,D_T)$, $T_j$ takes the value $|\{1\leq t\leq T:P_t\in\mathbf{I}_j\}|$ with probability one, and when $T_i=|\{1\leq t\leq T:P_t\in\mathbf{I}_j\}|$, $\mu_i^\pi(P_1,D_2,P_2,D_2,\ldots,P_T,D_T,T_j)=\nu_i^\pi(P_1,D_2,P_2,D_2,\ldots,P_T,D_T)$, the fifth identity holds since $D_t=f_0(P_t)+\varepsilon_t$, $\mathbb{E}_0^\pi[\varepsilon_t]=0$ and $P_t$ is independent of $\varepsilon_t$, and the sixth identity holds since $f_0$ and $f_j$ are only different in $\mathbf{I}_j$. Then (56) is obtained by combining (58) with (59).

### D.1. Proof of Lemma 3

The result for $j=0$ is trivial. When $1\leq j\leq J$, from the properties (1) and (2) of $S(x)$, $g_k(x)$ is infinitely differentiable. Now we check the property of $f_j^{(\mathfrak{b}(k))}(p)$.

$$
\begin{aligned}
\left|f_j^{\mathfrak{b}(k)}(p_1)-f_j^{\mathfrak{b}(k)}(p_2)\right| &= \left|\left(\frac{r_j(p_1)}{p_1}\right)^{(\mathfrak{b}(k))}-\left(\frac{r_j(p_2)}{p_2}\right)^{(\mathfrak{b}(k))}\right| \\
&= \left|\sum_{i=0}^{\mathfrak{b}(k)}\binom{\mathfrak{b}(k)}{i}\left(\frac{1}{p_1}\right)^{(\mathfrak{b}(k)-i)}r_j^{(i)}(p_1)-\sum_{i=0}^{\mathfrak{b}(k)}\binom{\mathfrak{b}(k)}{i}\left(\frac{1}{p_2}\right)^{(\mathfrak{b}(k)-i)}r_j^{(i)}(p_2)\right| \\
&= \left|\sum_{i=0}^{\mathfrak{b}(k)}\binom{\mathfrak{b}(k)}{i}(-1)^{(\mathfrak{b}(k)-i)}(\mathfrak{b}(k)-i)!\left(\frac{r_j^{(i)}(p_1)}{p_1^{\mathfrak{b}(k)-i+1}}-\frac{r_j^{(i)}(p_2)}{p_2^{\mathfrak{b}(k)-i+1}}\right)\right| \\
&\leq \left(\sum_{i=0}^{\mathfrak{b}(k)}\frac{\mathfrak{b}(k)!}{i!}\right)\max_{0\leq i\leq\mathfrak{b}(k)}\left|\frac{r_j^{(i)}(p_1)}{p_1^{\mathfrak{b}(k)-i+1}}-\frac{r_j^{(i)}(p_2)}{p_2^{\mathfrak{b}(k)-i+1}}\right|,
\end{aligned}
\tag{60}
$$

where the second identity follows the general Leibniz rule. Now we turn to the RHS of Eq. (60). In the following, we discuss in three cases: (1) $p_1,p_2\in\mathbf{I}_j$, (2) $p_1\in\mathbf{I}_j$ and $p_2\notin\mathbf{I}_j$, and (3) $p_1,p_2\notin\mathbf{I}_j$.

Case 1: $p_1 \in \mathbf{I}_j$ and $p_2 \in \mathbf{I}_j$.

If $0 \le i \le \mathfrak{b}(k) - 1$, we have

$$
\left| \frac{r_j^{(i)}(p_1)}{p_1^{\mathfrak{b}(k)-i+1}} - \frac{r_j^{(i)}(p_2)}{p_2^{\mathfrak{b}(k)-i+1}} \right|
$$

$$
\le \left| \frac{r_j^{(i)}(p_1)}{p_1^{\mathfrak{b}(k)-i+1}} - \frac{r_j^{(i)}(p_2)}{p_1^{\mathfrak{b}(k)-i+1}} \right| + \left| \frac{r_j^{(i)}(p_2)}{p_1^{\mathfrak{b}(k)-i+1}} - \frac{r_j^{(i)}(p_2)}{p_2^{\mathfrak{b}(k)-i+1}} \right|
$$

$$
\le \left| r_j^{(i)}(p_1) - r_j^{(i)}(p_2) \right| + \max_{p \in [1,2]} \left| r_j^{(i)}(p) \right| \left| \frac{p_1^{\mathfrak{b}(k)-i+1} - p_2^{\mathfrak{b}(k)-i+1}}{p_1^{\mathfrak{b}(k)-i+1} p_2^{\mathfrak{b}(k)-i+1}} \right|
$$

$$
\le \max_{p \in [1,2]} \left| r_j^{(i+1)}(p) \right| |p_1 - p_2| + \max_{p \in [1,2]} \left| r_j^{(i)}(p) \right| |p_1 - p_2| \left| \sum_{q=0}^{\mathfrak{b}(k)-i} p_1^q p_2^{\mathfrak{b}(k)-i-q} \right|
$$

$$
\le \eta(2J)^{i+1} \max_{x \in [0,2]} \left| g_k^{(i+1)}(x) \right| |p_1 - p_2| + \eta(2J)^i \max_{x \in [0,2]} \left| g_k^{(i)}(x) \right| |p_1 - p_2| (\mathfrak{b}(k) - i + 1) 2^{\mathfrak{b}(k)-i}
$$

$$
\le \left( \eta(2J)^{i+1} + (\mathfrak{b}(k) - i + 1)\eta(2J)^i 2^{\mathfrak{b}(k)-i} \right) |p_1 - p_2|
$$

$$
\le 2\eta(2J)^{i+1} |p_1 - p_2|, \tag{61}
$$

where the third inequality follows from mean value theorem, the existence of the $i+1$-th derivatives, $p_1 \ge 1$, $p_2 \ge 1$ and the fact that $p_1^{\mathfrak{b}(k)-i} - p_2^{\mathfrak{b}(k)-i+1} = (p_1 - p_2)(\sum_{q=0}^{\mathfrak{b}(k)-i+1} p_1^q p_2^{\mathfrak{b}(k)-i-q})$, the fourth inequality holds due to $p_1 \le 2$ and $p_2 \le 2$, the fifth inequality holds by our construction that $\max_{x \in [0,2]} \left| g_k^{(i)}(x) \right| \le 1$, the last inequality follows from $(\mathfrak{b}(k) + 1)2^{\mathfrak{b}(k)} \le 2J$. Then, for $i = \mathfrak{b}(k)$,

$$
\left| \frac{r_j^{(\mathfrak{b}(k))}(p_1)}{p_1} - \frac{r_j^{(\mathfrak{b}(k))}(p_2)}{p_2} \right|
$$

$$
\le \left| \frac{r_j^{(\mathfrak{b}(k))}(p_1)}{p_1} - \frac{r_j^{(\mathfrak{b}(k))}(p_2)}{p_1} \right| + \left| \frac{r_j^{(\mathfrak{b}(k))}(p_2)}{p_1} - \frac{r_j^{(\mathfrak{b}(k))}(p_2)}{p_2} \right|
$$

$$
\le \eta(2J)^{\mathfrak{b}(k)} \left| g^{(\mathfrak{b}(k))}(2J(p_1 - a_j)) - g^{(\mathfrak{b}(k))}(2J(p_2 - a_j)) \right| + |p_1 - p_2| \max_{p \in [1,2]} \left| r_j^{\mathfrak{b}(k)}(p) \right|
$$

$$
\le \eta(2J)^{\mathfrak{b}(k)} |2J(p_1 - p_2))|^{k-\mathfrak{b}(k)} + \eta(2J)^{\mathfrak{b}(k)} |p_1 - p_2|
$$

$$
\le \left( \eta(2J)^k + \eta(2J)^{\mathfrak{b}(k)} \right) |p_1 - p_2|^{k-\mathfrak{b}(k)}
$$

$$
\le 2\eta(2J)^k |p_1 - p_2|^{k-\mathfrak{b}(k)}, \tag{62}
$$

where the second inequality holds because $\left| \frac{r_j^{(\mathfrak{b}(k))}(p_2)}{p_1} - \frac{r_j^{(\mathfrak{b}(k))}(p_2)}{p_2} \right| \le$ $\left| \frac{1}{p_1} - \frac{1}{p_2} \right| \max_{p \in [1,2]} \left| r_j^{(\mathfrak{b}(k))}(p) \right| \le |p_1 - p_2| \max_{p \in [1,2]} \left| r_j^{(\mathfrak{b}(k))}(p) \right|$, the third inequality follows that $g^{(\mathfrak{b}(k))}(\cdot)$ is $(k - \mathfrak{b}(k))$-Hölder continuous, the fourth inequality holds because of

$|p_1 - p_2| \leq 1$ and $k - \mathfrak{b}(k) \leq 1$. Then, Eq. (60) can be simplified

$$\left| f_j^{\mathfrak{b}(k)}(p_1) - f_j^{\mathfrak{b}(k)}(p_2) \right|$$

$$\leq \left( \sum_{i=0}^{\mathfrak{b}(k)} \frac{\mathfrak{b}(k)!}{i!} \right) \max \left\{ \max_{0 \leq i \leq \mathfrak{b}(k)-1} 2\eta(2J)^{i+1} |p_1 - p_2|, 2\eta(2J)^k |p_1 - p_2|^{k-\mathfrak{b}(k)} \right\}$$

$$\leq \left( \sum_{i=0}^{\mathfrak{b}(k)} \frac{\mathfrak{b}(k)!}{i!} \right) \max \left\{ 2\eta(2J)^{\mathfrak{b}(k)} |p_1 - p_2|, 2\eta(2J)^k |p_1 - p_2|^{k-\mathfrak{b}(k)} \right\}$$

$$= \left( \sum_{i=0}^{\mathfrak{b}(k)} \frac{\mathfrak{b}(k)!}{i!} \right) \left( 2\eta(2J)^k |p_1 - p_2|^{k-\mathfrak{b}(k)} \right)$$

$$\leq \delta |p_1 - p_2|^{k-\mathfrak{b}(k)}, \tag{63}$$

where the last inequality holds due to $2\eta(2J)^k \leq 2\frac{1}{2^{3k+1}} \frac{1}{((\mathfrak{b}(k)+1)2^{\mathfrak{b}(k)})^k} \hat{\delta}^{\frac{1}{2k+1}} T^{-\frac{k}{2k+1}} (8(\mathfrak{b}(k) + 1)2^{\mathfrak{b}(k)} \hat{\delta}^{\frac{2}{2k+1}} T^{\frac{1}{2k+1}})^k = \hat{\delta}$.

Case 2: $p_1 \in \mathbf{I}_j$ and $p_2 \notin \mathbf{I}_j$.

Note that Eq. (61) still hold in this case. What we need to derive is the bound for $\left| \frac{r_j^{(\mathfrak{b}(k))}(p_1)}{p_1} - \frac{r_j^{(\mathfrak{b}(k))}(p_2)}{p_2} \right|$ (i.e., Eq. (62) can not be directly applied here). Define $p_2' := \mathrm{Proj}(p_2, \mathbf{I}_j)$. Note that $p_2'$ is either $a_j$ or $b_j$, $r_j(a_j) = r_j(b_j) = \frac{1}{2}\hat{\delta}$ and $r_j^{(i)}(a_j) = r_j^{(i)}(b_j) = 0$ for all $1 \leq i \leq \mathfrak{b}(k)$. Thus, we can have

$$\left| \frac{r_j^{(\mathfrak{b}(k))}(p_1)}{p_1} - \frac{r_j^{(\mathfrak{b}(k))}(p_2)}{p_2} \right| = \left| \frac{r_j^{(\mathfrak{b}(k))}(p_1)}{p_1} - \frac{r_j^{(\mathfrak{b}(k))}(p_2')}{p_2'} \right|$$

$$\leq 2\eta(2J)^k |p_1 - p_2'|^{k-\mathfrak{b}(k)} \leq 2\eta(2J)^k |p_1 - p_2|^{k-\mathfrak{b}(k)}, \tag{64}$$

where the first equality holds due to $r_j^{(\mathfrak{b}(k))}(p_2) = r_j^{(\mathfrak{b}(k))}(p_2') = 0$, the first inequality follows from Eq. (62) because $p_1$ and $p_2'$ are both in $\mathbf{I}_j$, and the second inequality holds due to the projection process. Together with Eqs. (60), (61) and (64), by the same calculation of Eq. (63), we can know $|f_j^{\mathfrak{b}(k)}(p_1) - f_j^{\mathfrak{b}(k)}(p_2)| \leq \delta |p_1 - p_2|^{k-\mathfrak{b}(k)}$.

Case 3: $p_1 \notin \mathbf{I}_j$ and $p_2 \notin \mathbf{I}_j$.

When $p_1$ and $p_2$ are not in $\mathbf{I}_j$, $r_j^{(i)}(p_1) = r_j^{(i)}(p_2) = 0$, for all $1 \leq i \leq \mathfrak{b}(k)$. From the first two lines of Eq. (60), we can have

$$\left| f_j^{\mathfrak{b}(k)}(p_1) - f_j^{\mathfrak{b}(k)}(p_2) \right| = \left| \frac{r_j(p_1)}{p_1^{\mathfrak{b}(k)+1}} - \frac{r_j(p_2)}{p_2^{\mathfrak{b}(k)+1}} \right|$$

$$\leq \frac{1}{2}\hat{\delta} \left| p_1^{\mathfrak{b}(k)+1} - p_2^{\mathfrak{b}(k)+1} \right|$$

$$\leq \frac{1}{2}\hat{\delta}|p_1 - p_2| \left| \sum_{q=0}^{\mathfrak{b}(k)} p_1^q p_2^{\mathfrak{b}(k)-q} \right|$$

$$\leq \frac{1}{2}\hat{\delta}(\mathfrak{b}(k) + 1)2^{\mathfrak{b}(k)}|p_1 - p_2|$$

$$\leq \delta|p_1 - p_2|^{k-\mathfrak{b}(k)},$$

where the third inequality holds due to $p_1 \leq 2$ and $p_2 \leq 2$, and the last inequality follows from $\hat{\delta} \leq \delta/((\mathfrak{b}(k) + 1)2^{\mathfrak{b}(k)} - 1)$.

Together with the above three cases, we draw the conclusion that for any $p_1, p_2 \in [1, 2]$, we have $|f_j^{\mathfrak{b}(k)}(p_1) - f_j^{\mathfrak{b}(k)}(p_2)| \leq \delta |p_1 - p_2|^{k-\mathfrak{b}(k)}$. We finish the proof.