# OpenReview forum: "Context-Based Dynamic Pricing with Partially Linear Demand Model"
_NeurIPS.cc/2022/Conference — NeurIPS 2022 Accept_

### Official Review · Reviewer_JfvX · 2022-07-03

**Rating:** 7
**Confidence:** 3
**Soundness:** 4 excellent
**Presentation:** 3 good
**Contribution:** 3 good

**Summary:**

This paper studies the contextual dynamic pricing problem under partially linear structural assumptions. In particular, the authors consider two demand models **DPLPE** and **DPLCE**. The former assumes a linear term in the price with an additive $\beta$-Holder continuous function of the context. The latter assumes a linear term in the context with an additive $k$-th order smooth function of the price.
The authors present two online algorithms and provide regret upper bound as well as lower bound guarantees for each of the models:
1. **DPLPE**:
   - upper bound of $\mathcal{O}\left(\sqrt{T} + \ln T\cdot T^{\frac{d}{d+2\beta}}\right)$.
   - lower bound of $\Omega\left(\max(\sqrt{T}, T^{\frac{d}{d+2\beta}})\right)$.
2. **DPLCE**:
   - upper bound of $\mathcal{O}\left(d\ln T(\sqrt{T} + (\delta T^{k+1})^{\frac{1}{2k+1}}))\right)$.
   - lower bound of $\Omega\left(\max(\sqrt{T},(\delta T^{k+1})^{\frac{1}{2k+1}}))\right)$.

**Questions:**

As this paper presents theoretical results I find the Numerical Study section redundant and would suggest providing more background on the proof techniques used in the paper.

**Limitations:**

I don't see any potential negative societal impact.

**Strengths And Weaknesses:**

**Strenghts**:
- The paper in general is well written and its contribution over previous works is presented clearly by the authors.
- The models considered in the paper are more general compared to known works and they provide new results on the problem of dynamic pricing.
- I find the algorithm design of both algorithms to be interesting and novel as it combines multiple ideas from previous works.
- The authors also provide lower bound guarantees which enhances the main results of the paper.

**Weaknesses**:
- As stated by the authors the lower bound in Theorem 4 is missing the dependence in $d$ which makes this lower bound less significant compared to the lower bound in Theorem 2.
- I believe that the problem setup can be presented more clearly for the reader with better formatting.
- I feel like the main text could benefit from a more comprehensive technical overview.

---

> ### Author Response · Authors · 2022-08-02
> **Response to Reviewer JfvX**
>
> Thank you for your insightful comments and valuable feedback. We next provide our detailed response to your questions and comments.
>
> In this work, we focus on working on the dependence on $T$ in Theorem 4, since in a real-world application, the dimension of context $d$ is relatively very small comparing with $T$.  Admittedly, matching the upper bound with the lower bound in terms of the dependency of dimension $d$ is important, especially when $d$ is very large. This question is indeed one of the important future research directions, and we are working along this line.
>
> For the numerical study, we want to provide some more insights on the impact of parameters like $\beta$, $\delta$ from the empirical side, because they are hardly discussed in the previous literature. We will shorten the numerical part to make it more compact.
>
> We feel really sorry that we do not provide enough technical details due to the space limit. We will highlight more technical details in our revised version as follows.
>
> 1. For Theorem 1, we will modify the paragraph right after the theorem (from line 165 in the original version) as following:
> The bound in Theorem 1 consists of two terms $\widetilde{\mathcal{O}}\left(\sqrt T\right)$ and $\widetilde{\mathcal{O}}\left(T^\frac{d}{d+2\beta}\right)$. The first term $\widetilde{\mathcal{O}}\left(\sqrt T\right)$  is caused by the random shocks that we introduce to estimate b, capturing the complexity of learning b. …… The second term $\widetilde{\mathcal{O}}\left(T^\frac{d}{d+2\beta}\right)$ is incurred by our inaccuracy of approximating the non-parametric function $g\left(\cdot\right)$ with a constant in each small bin.
>
> 2. For Theorem 2, we have highlighted the main idea and the techniques that we use as following:
> To prove the second lower bound $\widetilde{\Omega}\left(T^\frac{d}{d+2\beta}\right)$, we construct a series of Hölder continuous functions in $\left[0,1\right]^d$ that are difficult to distinguish from each other. We then apply the Bretagnolle–Huber inequality (see [6]) and KL divergence arguments to bound the regret of any algorithm from below.
>
> 3. For Theorem 3, we will highlight the technical background as following just after Theorem 3:
> By the idea of optimism over OFU, we can upper bound the regret by the length of the confidence interval and then applied the well-known concentration inequality from [A] to well control the growth of the confidence interval.
>
> 4. For Theorem 4, we have mentioned the main idea as following: The proof of the second lower bound $\Omega((\delta T^{k+1})^{\frac{1}{2k+1}})$ relies on constructing a series of $k$th-order smooth functions with parameter $\delta$ and applying Pinsker's inequality to bound the total variation of two probability measures through their KL divergence.
>
> [A] Abbasi-Yadkori, Y., Pál, D., Szepesvári, C. (2011). Improved algorithms for linear stochastic bandits. Advances in neural information processing systems, 24.
>
> Thanks to the reviewer again for raising these important points and giving us a chance to further improve our paper’s flow.

---

### Official Review · Reviewer_pEPr · 2022-07-03

**Rating:** 5
**Confidence:** 4
**Soundness:** 3 good
**Presentation:** 3 good
**Contribution:** 2 fair

**Summary:**

This paper studies the context-based dynamic pricing, where the unknown expected demand admits a semi-parametric partially linear structure. Two special cases of semi-parametric partially linear models (Linear Pricing Effect and Linear Contextual Effect) are considered. Two new algorithms (DPLPE and DPLCE) are proposed and their regret upper bounds and matching lower bounds are established.


**Questions:**

(1) It is important to discuss the technical novelty of this paper beyond existing semi-parametric dynamic pricing papers.

(2) It is important to provide some guidance on how to choose the unknown parameters (bounds of $b$, $\beta$, etc) in a data-driven way. It is also helpful to study the sensitivity of the choice of these parameters in experiments.

(3) It is helpful to list all assumptions used for the theorems. When discussed with regret bounds in the literature, it is important to also mention the difference in the assumptions.

(4) It is important to study the model misspecification case, e.g., when the true model is linear or purely nonparametric, in the experiments.




**Ethics Review Area:**

["I don’t know"]

**Limitations:**

No potential negative societal impact was discussed in the checklist. No limitation was discussed in the paper.

**Strengths And Weaknesses:**

Strengths

(1) In spite of a theoretical paper, it is well written and easy to follow.

(2) The proved matching lower bound is useful to understand the limit of the considered problem.


Weaknesses

(1) Technical novelty: The proposed partially linear demand model is a natural extension of existing linear demand model and nonparametric demand model. On the other hand, a similar partially linear demand model has been proposed in dynamic pricing literature, see below. None of them was mentioned in the paper. In fact, these papers consider a binary choice model which is arguably more challenging than the one considered in this paper.

Jianqing Fan, Yongyi Guo, Mengxin Yu (2021), Policy Optimization Using Semi-parametric Models for Dynamic Pricing, https://arxiv.org/abs/2109.06368.

Jianyu Xu, Yu-Xiang Wang (2022), Towards Agnostic Feature-based Dynamic Pricing: Linear Policies vs Linear Valuation with Unknown Noise, AISTATS 2022.

(2) The proposed algorithm and theoretical analysis require the knowledge of some true parameters, e.g., the upper and lower bounds of price, the upper and lower bounds of the true price coefficient, The continuous parameter $\beta$ of the unknown function $g$. In practical pricing applications, knowing upper and lower bounds of price is arguable and a mild assumption. But it is less justifiable to know the bound of $b$ and $\beta$ in practice.

(3) It is unclear what assumptions were assumed in the main Theorems (Theorems 1-4). In addition, when the authors compare the proved regret bounds with those in the literature, it is also important to discuss if their model assumptions are comparable. It would be more convincing if the faster regret bound is not obtained under much stronger assumptions.

(4) The experiments of this paper only study the performance when the true model is the proposed partially linear demand model. No model misspecification was studied. It is unclear if it is always safe to use the proposed algorithm in practical pricing applications.

~~~~~~~~~~~~~
After rebuttal: Thanks the authors for carefully addressing all my concerns. My last three comments have been nicely addressed. I have raised my rating to Borderline Accept. On the other hand, while I understand the difference (e.g., nonparametric on noise v.s. nonparametric on covariates; binary feedback v.s. continuous feedback) of this paper compared to existing pricing literature with partial linear demand model, I am not fully convinced by the "technical novelty" beyond them. So I choose to only increase the rating to Borderline Accept.

---

> ### Author Response · Authors · 2022-08-02
> **Response to Reviewer pEPr -- Part I**
>
> Thank you for your insightful comments and valuable feedback. We next provide our detailed response to your questions and comments.
>
> 1. Technical novelty comparing with binary choice model
>
> Thank you very much for pointing out the two important papers [A] [B], and we find [C] is also closely related in this stream. We are sorry about missing these works and have read them with much carefulness. Below we give a detailed comparison between these works with ours. We will carefully add this discussion to our related work and Table 1 in our revised version.
>
> As mentioned by the reviewer, [A] [B] and [C] are based on a binary choice model with unknown noise distribution and linear valuation function of customer. Specifically, every customer buys the product with probability $1-F\left(p_t-\theta^\top x_t\right)$, where $F$ is the CDF of random noise. [A] [B] and [C] consider a very important question of how to estimate \theta without knowing $F$. [A] propose an impressive estimator of \theta in their Eq. (3.1), [B] integrate the idea of discretization and EXP4 algorithm cleverly, and [C] reduce the challenge to a logistics regression. Conceptually, one can transform our demand model into the one where each customer buys the product with probability $bp_t+g\left(x_t\right)$ (for DPLPE) or $f\left(p_t\right)+a^\top x_t$ (for DPLCE), assuming $bp_t+g\left(x_t\right)$  and $f\left(p_t\right)+a^\top x_t$ fall in $\left[0,1\right]$.  Although the unknown noise distribution is not our challenge as in [A], [B] and [C], we still need to estimate $b$ and the non-parametric function $g\left(\cdot\right)$ at the same time for DPLPE (or a and the non-parametric function $f\left(\cdot\right)$ for DPLCE). Despite that we have an additive structure, it is still challenging to decouple $bp_t$ and  $g\left(x_t\right)$ for DPLPE (or $f\left(p_t\right)$ and $a^\top x_t$ for DPLCE) with only a buying/not-buying feedback. In short words, [A] [B] and [C] need to estimate $F\left(\cdot\right)$ and $\theta$ with only one feedback, whereas we need to estimate $b$ and $g\left(\cdot\right)$ (or $a$ and $f\left(\cdot\right)$) with only one feedback. We completely agree that [A] [B] and [C] consider an important and challenging question, but also would like to point out that our models are faced with several different challenges. In our revised version, we will point out these three important papers and compare these works with ours carefully.
>
> Furthermore, the binary model and our model do not contradict with each other. There are several middle grounds between these two models, which may also be of our interest. An example is in the binary model, $F\left(\cdot\right)$ is known but the valuation function is non-parametric, i.e., $g\left(x_t\right)$ as in our model, or even $F\left(\cdot\right)$ is unknown and the valuation function is non-parametric.
>
> [A] Jianqing Fan, Yongyi Guo, Mengxin Yu (2021), Policy Optimization Using Semi-parametric Models for Dynamic Pricing, https://arxiv.org/abs/2109.06368.
>
> [B] Jianyu Xu, Yu-Xiang Wang (2022), Towards Agnostic Feature-based Dynamic Pricing: Linear Policies vs Linear Valuation with Unknown Noise, AISTATS 2022.
>
> [C] Luo, Y., Sun, W. W. (2021). Distribution-free contextual dynamic pricing. arXiv preprint arXiv:2109.07340.

---

> ### Author Response · Authors · 2022-08-02
> **Response to Reviewer pEPr -- Part II**
>
> 2. The choice of the unknown parameter.
>
> 2.1. The bounds of $b$ and $\left|\left|a\right|\right|$. We admit that the knowledge of the bounds of $b$ and $||a||$ is a strong assumption and we will make this claim in the revised version, although it is a common assumption in the dynamic pricing literature (see, e.g.,  [D], [E], [F], [J]). There are already several empirical methods to estimate the bound of $b$ from the historical data. [F] provides an efficient data-driven approach based on two-stage least squares (2SLS) method with the careful choice of the instrument variable, together with a case study of the company, Oracle Retail. Their estimates of $b$ for different categories of fashion items were found to be of the same order of magnitude, lying in the range $[-1,-0.1]$. [G] also propose some useful tools and they analyze $1851$ price elasticities across different markets. They observe a mean price elasticity of $-2.62$ with $50$ percent of the observations between $-1$ and $-3$. We will add the discussion to our revised version. For the norm of $a$, we can repeat the same process.
>
> 2.2. The choices of $\beta$ and $k$. We will make it clear in the revised version that $\beta$ and $k$ need to be known in advance and will point out explicitly that this is one of the limitations of our work. Although most of the existing literature assume the knowledge of smoothness parameters (see, e.g., [H]), we completely agree that the exact values of these parameters are not available in practice. When there is no such prior knowledge, we can estimate them from the historical data. For example, Section G of [A] provides a cross-validation method to determine the smoothness parameters, which can also be applied to our setting after slight modifications. Also, we can do the grid-search for \beta and k, and find the ones that fit the historical data best by solving regression using polynomial functions (i.e., $c_1{\left||x\right||}^{\ \beta}+c_2$ and $c_3p^k+c_4p^{\left\lfloor k\right\rfloor}+\ldots+c_{\left\lfloor k\right\rfloor+1}$ respectively).
>
> We also would like to point out that in a recent paper [I], the authors establish a very important negative result that designing algorithms that adapt to unknown smoothness of payoff functions is generally impossible. This implies that in our setting, if we do not have any knowledge about $\beta$ and $k$, it is virtually impossible to design an efficient algorithm. [I] point out that if an extra crucial self-similarity assumption holds, it is possible to design efficient algorithm for bandit with discrete arms. However, whether the self-similarity condition can hold in our dynamic pricing setting remains unclear, and how to extend [I]'s algorithms to continuous price space is beyond the scope of this paper. Nevertheless, we believe that this is an important future research direction, and will provide more discussions on this issue in our conclusion section.
>
> [D] Besbes, O.,  Zeevi, A. (2015). On the (surprising) sufficiency of linear models for dynamic pricing with demand learning. Management Science, 61(4), 723-739.
>
> [E] Miao, S., Chen, X., Chao, X., Liu, J., Zhang, Y. (2022). Context‐based dynamic pricing with online clustering. Production and Operations Management.
>
> [F] Nambiar, M., Simchi-Levi, D.,  Wang, H. (2019). Dynamic learning and pricing with model misspecification. Management Science, 65(11), 4980-5000.
>
> [G] Bijmolt, T. H., Van Heerde, H. J., Pieters, R. G. (2005). New empirical generalizations on the determinants of price elasticity. Journal of marketing research, 42(2), 141-156.
>
> [H] Hu, Y., Kallus, N.,  Mao, X. (2020). Smooth contextual bandits: Bridging the parametric and non-differentiable regret regimes. In Conference on Learning Theory (pp. 2007-2010). PMLR.
>
> [I] Gur, Y., Momeni, A.,  Wager, S. (2022). Smoothness-adaptive contextual bandits. Operations Research.
>
> [J] Keskin, N. B., Zeevi, A. (2014). Dynamic pricing with an unknown demand model: Asymptotically optimal semi-myopic policies. Operations research, 62(5), 1142-1167.

---

> ### Author Response · Authors · 2022-08-02
> **Response to Reviewer pEPr -- Part III**
>
> 3. Assumptions in main theorems.
>
> We’d thank the reviewer for this suggestion. We are sorry that we did not make the assumptions clear enough in Theorem 1 and Theorem 3, which might cause some confusion. In our revised version, we will state our assumptions of demand functions directly in Theorem 1 (i.e., $\beta$-Hölder continuity of $g\left(x\right)$  and boundedness of $b$) and Theorem 3 (i.e., $k$-th order smoothness of $f\left(p\right)$ and boundedness of $a$). For the lower bounds in Theorem 2 and Theorem 4, all the assumptions have already been explicitly presented in the “sup” environment.
>
> In the current version, when comparing our results with those in the existing literature, we have explicitly pointed out the differences of their assumptions and ours. Specifically, in the paragraph after Theorem 2, we have carefully discussed the difference in comparison with [23] and [8]. [23] consider almost the same model as ours but have a completely different benchmark when defining the regret. [8] do not assume the separable structure and only assume Lipschitz continuity instead of  $\beta$-Hölder continuity. Thus, when $\beta=1$, our model is comparable with theirs. For this Lipschitz continuous model, we have shown the improvement of our regret bounds due to the separable structure. For a more detailed discussion, please refer to lines 186-199 in our original version. In the paragraph after Theorem 3, we have also shown the difference in assumptions between ours and that in [30] and discussed how our general model can be reduced to [30]. The details can be seen in lines 251-263. As a summary, we have also listed the key assumptions of demand models in the existing studies and our work in our Table 1.
>
> In addition, we would like to mention that for DPLCE, even under a much stronger assumption that $a$ is exactly known, our regret bound is not improvable in terms of the dependency of $T$ and $\delta$. Thank you again for your valuable suggestions, which make the presentation and exposition of our paper clearer.
>
> 4. Model misspecification issue.
>
> Thank you for raising the important issue of model misspecification, which is very crucial especially in the real-world applications.
>
> When the true demand is linear, both algorithms ADPLP and ADPLC can lead to a sublinear regret theoretically. Specifically, ADPLP will have a regret upper bound in the order $\widetilde{\mathcal{O}}\left(\sqrt T\vee T^\frac{d}{d+2\beta}\right)$ for any input $\beta$, since linear function is $\beta$-Hölder continuous for any $\beta\in\left(0,1\right]$. When $d=1$ and we choose $\beta\geq\frac{1}{2}$, we can even obtain the optimal regret $\widetilde{\Theta}\left(\sqrt T\right)$. Under other settings, though not optimal, ADPLP still eventually converges to the optimal solution. Since ADPLP does not have the prior information of the linear structure, the sublinear regret is the best we can expect. For ADPLC, since linear functions are $k$th-order smooth, the regret upper bound can be guaranteed by Theorem 3. If we have more information, choosing $k\geq 3$ and $\delta=0$ or $k=ln{T}$, we can again obtain the optimal $\widetilde{\Theta}\left(\sqrt T\right)$ regret. Therefore, when the true demand is linear, both algorithms are robust to the model misspecification.
>
> When the true demand is purely non-parametric, the current regret notion is no longer a reasonable measure of the algorithm’s performance. Similar to the discussion in [F], we cannot expect a sublinear regret if the benchmark is the true optimal policy. In fact, how to define a reasonable measure under the mis-specified case is a fundamental question that is worthy of a separate study. Therefore, in the pure non-parametric setting, even if one can find some problem instances for which our algorithms ADPLP and ADPLC perform well and achieve a sublinear regret, these problem instances may not be representative enough. Considering that a thorough study of this issue may be out of the scope of this paper, we prefer to leave it to future research. We hope that this treatment is fine with you.
>
> 5. Limitation of this paper.
>
> The main limitation we discussed in our paper is that Theorem 3 and Theorem 4 do not match with respect to dimension d (see the paragraph just after Theorem 4). In the revised version, we will discuss more about the limitation due to the need of the bounds on $b$, $\beta$ and $k$, as pointed out by the reviewer. We'd thank the reviewer for giving us a good opportunity to rethink the limitation of our work.

---

### Official Review · Reviewer_1rnn · 2022-07-04

**Rating:** 7
**Confidence:** 5
**Soundness:** 4 excellent
**Presentation:** 4 excellent
**Contribution:** 3 good

**Summary:**

This paper studies two online contextual dynamic pricing problem settings: DPLPE where demand is linear wrt price, and DPLCE where demand is linear wrt context. For DPLPE problem, they assume the realized demand $D_t = bp_t+g(x_t) + \epsilon_t$ with $g(x)$ being $\beta$-Holder continuous and $\epsilon_t$ being a subGaussian noise. They propose an ADPLP algorithm that adopts a space binning and a random shock techniques. With ADPLP, they achieve an optimal $\tilde{O}(\sqrt{T}\vee T^{\frac{d}{d+2\beta}})$ regret up to logarithmic factors by proving both the upper and the lower regret bounds, where $d$ is the dimentionality of features. For DPLCE problem, they assume the realized demand $D_t = f(p_t) +a^{\top}x_t + \epsilon_t$, with $f(p)$ being $k^{\text{th}}$-order smooth with a small parameter $\delta$ that could be $T$-dependent. They present an ADPLC algorithm that adopts a local polynomial approximation and a biased linear contextual bandit with an optimistic OFU strategies. With ADPLC, they achieve an optimal $\tilde{O}_d(\sqrt{T}\vee(\delta T^{k+1})^{\frac1{2k+1}})$ regret up to logarithmic factors, by proving both the upper and the lower regret bounds. Finally, they conduct numerical experiments and show that the simulation results of ADPLP and ADPLC ourperform all benchmarks.

**Questions:**

Questions:

(1) See the related work issue I mentioned above. Maybe a good idea is to list these key properties/assumptions in categories (like what they did in [30] Table 1).

(2) Notice that ADPLC require knowledge on the noise std $\sigma$. Does ADPLP require it as well? It is sometimes decisive to assume a parametric or non-parametric noise in an online learning problem, so I suggest the authors to specify these properties in detail, and briefly discussion how they contribute to your analysis and results.

Minor suggestions:

(1) Avoid notation repeatance: e.g., $d$ for feature dimension and also for expected demand.

(2) A few typos: e.g. line 42: $x_t\in[0,1]^d$.


**Limitations:**

Limitations and potential extensions of this work are well discussioned.
There is no discussions on social impact as it is a work of theory. However, I indeed suggest the authors to consider any potential ethic issue that might occur in a pricing problem with these assumptions you have specified.

**Strengths And Weaknesses:**

Strengths:

(1) This work generalizes the problem settings of both linear demand and linear context problems. For DPLPE, they consider $\beta$-Holder class instead of only Lipschitz that was broadly assumed by previous works, and they improve the comparison by replacing the linear benchmark with the true demand function in [8] and [23]. For DPLCE, they consider $k$-th-order smooth with not only integer but also non-integer $k$'s, and they also emphasize the role that $\delta$ plays instead of treating it as a constant as previous works did.

(2) For both DPLPE and DPLCE, they design algorithms with provable optimal regret bounds. These results are significant as they not only match the order of $T$ but also those of $\beta$ and $\delta$. Moreover, for DPLCE, their upper and lower bound match those in [30], indicating that a linear context added on the demand curve might not require substantially more information to learn.

(3) Their numerical experiments are comprehensive and the results are well-displayed.

Weaknesses:

(1) Some related literatures should be discussed with more details. For example, the stream of binary demand model. As this work also assume a noisy feedback, the only difference of binary feedback is that the noise distribution is dependent on $p$ and $x$ while this work assumes iid. Since the closely-related work [30] also assumes a binary feedback, the results of this work do not actually cover those in [30]. Overall, the binary feedback is an important property in many pricing problem settings, and I suggest the authors to be aware of this issue and place this paper in the related literatures with more precision.

---

> ### Author Response · Authors · 2022-08-02
> **Response to Reviewer 1rnn**
>
> Thank you for your insightful comments and valuable feedbacks. We next provide our detailed response to your questions and comments.
>
> 1. Binary Demand structure.
>
> We really want to thank the reviewer for bringing this important issue up. We first want to mention that the demand in [30] ([31] in the revised version) is not binary, but is a continuous variable in [0,1]. Below we give a detailed comparison between the binary demand structure with ours. We will also discuss carefully in our related work and Table 1.
>
> In the binary model, the customer buys the product with probability $1-F\left(p_t-\theta^\top x_t\right)$, where $F$ is the CDF of the random noise of the customer’s valuation. This stream of works starts from the case where F is well known in advance (see, [A], [B] and [C]). The knowledge of $F\left(\cdot\right)$  reduces the complexity of solving the problem since $\theta$ is the only parameters that need estimating. Recently, [D] [E] and [F] consider a very important question of how to estimate $\theta$ without knowing $F$. [D] propose an impressive estimator of $\theta$ in their Eq. (3.1). [E] integrate the idea of discretization and EXP4 algorithm cleverly and [F] reduce the challenge to a logistics regression.  In our model, the distribution of random shock makes no difference to the optimal pricing strategy. Thus, whether the distribution is parametric/known or not is not important in our formulation. As pointed out by [G], the assumption about parametric and non-parametric noise is indeed decisive for the binary reward setting.
>
> In order to compare with the binary choice model more clearly, conceptually, one can transform our demand model into the one where each customer buys the product with probability $bp_t+g(x_t)$ (for DPLPE) or $f(p_t)+a^\top x_t$ (for DPLCE), assuming $bp_t+g(x_t)$  and $f(p_t)+a^\top x_t$ fall in $[0,1]$.  Although the unknown $F$ is not our challenge as [D] [E] and [F], we still need to estimate b and the non-parametric function $g(\cdot)$ (or $a$ and the non-parametric function $f\left(\cdot\right)$) at the same time. Despite that we have an additive structure on the price effect and the context effect, it is still challenging to decouple $bp_t$ and  $g\left(x_t\right)$  (or $f\left(p_t\right)$ and $a^\top x_t$) with only a buying/not-buying feedback from the customer. In short words, [D] [E] and [F] need to estimate $F\left(\cdot\right)$ and $\theta$ with only one feedback and we need to estimate $b$ and $g\left(\cdot\right)$ (or $a$ and $f\left(\cdot\right)$) with only one feedback.
>
> Furthermore, the binary model and our model do not contradict with each other. There are several middle grounds between these two models, which are of our great interest and we also list as our future work. An example is in the binary model what if $F\left(\cdot\right)$ is known but the valuation function is non-parametric i.e., $g\left(x_t\right)$ in our model, and even what if $F\left(\cdot\right)$ is unknown and the valuation function is non-parametric.
>
> [A] Xu, J.,  Wang, Y. X. (2021). Logarithmic regret in feature-based dynamic pricing. Advances in Neural Information Processing Systems, 34, 13898-13910.
>
> [B] Javanmard, A.,  Nazerzadeh, H. (2019). Dynamic pricing in high-dimensions. The Journal of Machine Learning Research, 20(1), 315-363.
>
> [C] Javanmard, A. (2017). Perishability of data: dynamic pricing under varying-coefficient models. The Journal of Machine Learning Research, 18(1), 1714-1744.
>
> [D]Jianqing Fan, Yongyi Guo, Mengxin Yu (2021), Policy Optimization Using Semi-parametric Models for Dynamic Pricing, https://arxiv.org/abs/2109.06368.
>
> [E]Jianyu Xu, Yu-Xiang Wang (2022), Towards Agnostic Feature-based Dynamic Pricing: Linear Policies vs Linear Valuation with Unknown Noise, AISTATS 2022.
>
> [F] Luo, Y., Sun, W. W. (2021). Distribution-free contextual dynamic pricing. arXiv preprint arXiv:2109.07340.
>
> [G] Wang, H., Talluri, K., Li, X. (2021). On Dynamic Pricing with Covariates. arXiv preprint arXiv:2112.13254.
>
> 2. Knowledge of the proxy variance of the noise ${\sigma}^{2}$
>
> Thank you for carefully reading our paper and raising this question. You are completely right that in ADPLC, the variance proxy $\sigma^2$ needs to be known as an input to our algorithm. In ADPLP, we do not need to know $\sigma^2$. In the revised version, we will clearly mention this point. As in our response to the first question, the assumption of parametric or non-parametric noise distribution is not essential in our model, but is indeed decisive for the binary reward setting. In the revised version, we will make it clear about the difference between these two modelling approaches and discuss more about the roles of the noise parameter and distribution in our setting.
>
> 3. Notation and typos.
>
> We are sorry about our carelessness. We will double check our paper carefully and fix the typos including the one mentioned by the reviewer. We will also change the notation for expected demand to $\mu$.

---

> > ### Comment · Reviewer_1rnn · 2022-08-04
> > **Thanks for the authors' response!**
> >
> > The authors' response clarifies all of the points that I did not understand. As for [30] (now [31]), I tended to say that it covers the binary-feedback setting (as they assume the realized demand "random" instead of "noisy"), and I'm sorry for the confusing. The authors' explanation are informative, and I'm glad to see that you have already included it in your discussions on related works.
> >
> > I'm pretty sure that this work should get in. However, I'm not sure whether I should raise my score since there is a recalibration on the grades this year. I'll make necessary changes as soon as this gets clearer.

---

> > > ### Author Response · Authors · 2022-08-08
> > > **Thanks for the reviewer's time!**
> > >
> > > We really appreciate the reviewer’s insightful comments that help our paper become stronger. We have learned quite a lot from the reviewer’s professionalism and patience. Thank you so much for your valuable time.

---

### Official Review · Reviewer_JA1m · 2022-07-07

**Rating:** 6
**Confidence:** 4
**Soundness:** 3 good
**Presentation:** 4 excellent
**Contribution:** 3 good

**Summary:**

The paper studies an online learning and contextual pricing problem with semi-parametric partially linear demand models, where the demand function is the sum of a linear function with an unknown coefficient(s) of price (or context, resp.) and an unknown function of the context (or price, resp.). For the above two demand models, the paper develops two corresponding online pricing algorithms with provable regret upper bounds and matching lower bounds with respect to the horizon $T$.

**Questions:**

I hope the author can address in the rebuttal the two points mentioned in the weakness above and I may adjust my final rating based on the response.

**Limitations:**

NA.

**Strengths And Weaknesses:**

Strengths:
Overall, the paper is well-written.
Formulation-wise, the paper generalizes the commonly used Lipschitz continuous assumption to Holder continuity. Importantly, the results show that the additional structure of linearity indeed improves the regret bound.
Algorithm-wise, the paper combines several online learning techniques, including binning and approximation, random shock in pricing, and upper confidence bound algorithm, so as to estimate the non-parametric part of the demand functions.
Analysis-wise, the paper gives a comprehensive analysis of the statistical complexity/regret of the formulated problems.

Weakness:
 1. In Algorithm 2, when solving the UCB optimization problem in line 10, what is its complexity? And in general how to solve it (efficiently)?
 2. The algorithms seem to require the knowledge of $\beta$ and $k$, the authors should state these explicitly in the assumptions and discuss how to estimate these quantities when there is no such prior knowledge.

Minor comment:
Upon reviewing the literature, the paper misses one stream of works related to semi-parametric dynamic pricing (see below). The author should discuss the contribution against these works.
- Policy optimization using semiparametric models for dynamic pricing (Fan et al. 2021)
- Distribution-free contextual dynamic pricing (Luo and Sun, 2021)
- Towards agnostic feature-based dynamic pricing: Linear policies vs linear valuation with unknown noise (Xu and Wang, 2022)

Also, the authors should mention that the matching of lower bounds refers to the dependency on $T$ but not with respect to $d$. As in the discussion of Theorem 4, one of the lower bounds doesn't meet the corresponding upper bound in the context dimension.

---

> ### Author Response · Authors · 2022-08-02
> **Response to Reviewer JA1m -- Part I**
>
> Thank you for your insightful comments and valuable feedback. We next provide our detailed response to your questions and comments.
>
> 1. Complexity of solving the optimization problem in Algorithm 2 (line 10).
>
> Thank you for raising this important question. We have carefully looked into the computational issue of our algorithm. We would like to point out that the objective function in lines 10-11 of our Algorithm 2 is generally non-concave. Note that even in the setting where the demand is linear with price and there is no context, such an optimization problem is not concave (see Figure 1 in [C]). Nevertheless, the optimization problem in lines 10-11 of our Algorithm 2 is simply a univariate optimization problem (i.e., the price $p$ is the only decision variable). Since $\langle \hat\theta_{t,i} , \varphi(p)\rangle$ and $\phi(p,x_t)^\top V_{t,i}^{-1}\phi(p,x_t)$ are both polynomial functions of $p$ with explicit expressions, the objective function is easily evaluated. Therefore, we can discretize the $\textbf{I}_i$ to a satisfying granularity and compare the objective function at each discretized point. Besides, since $N$ is relatively large, the length of $\textbf{I}_i$ is relatively small, which can empirically benefit the speed of solving the optimization problem.
>
> We have also investigated the computational issues of those algorithms based on OFU principle in the existing literature. In fact, for almost all the algorithms based on OFU principle with a continuous action space, there exist similar computational challenges (see, e.g., [A], [B]). Involving a high-dimensional action space, the algorithm in [A] can be even more difficult to compute. [B] need to solve almost exactly the same optimization problem as our algorithm, but do not provide a careful discussion about the complexity. Thus, in general, we are not expecting an efficient algorithm theoretically and believe that the computational complexity of the algorithms based on OFU is a common challenge in the literature.
>
> [A] Abbasi-Yadkori, Y., Pál, D., Szepesvári, C. (2011). Improved algorithms for linear stochastic bandits. Advances in neural information processing systems, 24.
>
> [B] Wang Y, Chen B, Simchi-Levi D (2021) Multimodal dynamic pricing. Management Science 67(10):6136–6152.
>
> [C] Bu, J., Simchi-Levi, D.,  Xu, Y. (2022). Online pricing with offline data: Phase transition and inverse square law. Forthcoming in Management Science.
>
> 2. The knowledge of $\beta$ and $k$.
>
> Thank you for reminding us of the unknown parameters $\beta$ and $k$. We will make it clear in the revised version that $\beta$ and $k$ need to be known in advance and will point out explicitly that this is one of the limitations of our work. Although most of the existing literature assume the knowledge of smoothness parameters (see, e.g., [B] and [D]), we completely agree that the exact values of these parameters are not available in practice. When there is no such prior knowledge, we can estimate them from the historical data. For example, Section G of [F] provides a cross-validation method to determine the smoothness parameters, which can also be applied to our setting after slight modifications. Also, we can do the grid-search for $\beta$ and $k$, and find the ones that fit the historical data best by solving regression using polynomial functions (i.e., $c_1{\left||x\right||}^{\ \beta}+c_2$ and $c_3p^k+c_4p^{\left\lfloor k\right\rfloor}+\ldots+c_{\left\lfloor k\right\rfloor+1}$ respectively).
>
> We also would like to point out that in a recent paper [E], the authors establish a very important negative result that designing algorithms that adapt to unknown smoothness of payoff functions is generally impossible. This implies that in our setting, if we do not have any knowledge about $\beta$ and $k$, it is virtually impossible to design an efficient algorithm. [E] point out that if an extra crucial self-similarity assumption holds, it is possible to design efficient algorithm for bandit with discrete arms. However, whether the self-similarity condition can hold in our dynamic pricing setting remains unclear, and how to extend [E]'s algorithms to continuous price space is beyond the scope of this paper. Nevertheless, we believe that this is an important future research direction, and will provide more discussions on this issue in our conclusion section.
>
> [D] Hu, Y., Kallus, N.,  Mao, X. (2020). Smooth contextual bandits: Bridging the parametric and non-differentiable regret regimes. In Conference on Learning Theory (pp. 2007-2010). PMLR.
>
> [E] Gur, Y., Momeni, A.,  Wager, S. (2022). Smoothness-adaptive contextual bandits. Operations Research.
>
> [F] Jianqing Fan, Yongyi Guo, Mengxin Yu (2021), Policy Optimization Using Semi-parametric Models for Dynamic Pricing, https://arxiv.org/abs/2109.06368.
>
> (To be continued...)

---

> ### Author Response · Authors · 2022-08-02
> **Response to Reviewer JA1m -- Part II**
>
> 3. One missing stream of related works.
>
> Thank you very much for pointing out the three important papers. We are sorry about missing these works and have read them with much carefulness. Below we give a detailed comparison between these works with ours, and we will add these discussions to the related work and Table 1 in our revised version. As mentioned by the reviewer, [F], [G] and [H] are based on a binary choice model with unknown noise distribution and linear valuation function of customer. Specifically,  every customer buys the product with probability $1-F\left(p_t-\theta^\top x_t\right)$, where $F$ is the CDF of random noise. [F], [G] and [H] consider a very important question of how to estimate $\theta$ without knowing $F$. [F] propose an impressive estimator of $\theta$ in their Eq. (3.1), [G] integrate the idea of discretization and EXP4 algorithm cleverly, and [H] reduce the challenge to a logistics regression. Conceptually, one can transform our demand model into the one where each customer buys the product with probability $bp_t+g\left(x_t\right)$ (for DPLPE) or $f\left(p_t\right)+a^\top x_t $ (for DPLCE), assuming $bp_t+g\left(x_t\right)$  and $f\left(p_t\right)+a^\top x_t$ fall in $\left[0,1\right]$.  Although the unknown noise distribution is not our challenge as in [F], [G] and [h], we still need to estimate $b$ and the non-parametric function $g\left(\cdot\right)$ at the same time for DPLPE (or a and the non-parametric function $f\left(\cdot\right)$ for DPLCE). Despite that we have an additive structure, it is still challenging to decouple $bp_t$ and  $g\left(x_t\right)$ for DPLPE (or $f\left(p_t\right)$ and $a^\top x_t$ for DPLCE) with only a buying/not-buying feedback. In short words, [F], [G]and [H] need to estimate $F\left(\cdot\right)$ and $\theta$ with only one feedback, whereas we need to estimate $b$ and $g\left(\cdot\right)$ (or $a$ and $f\left(\cdot\right)$) with only one feedback. We completely agree that [F], [G] and [H] consider an important and challenging question, but also would like to point out that our models are faced with several different challenges. In our revised version, we will point out these three important papers and compare these works with ours carefully.
>
> Furthermore, the binary model and our model do not contradict with each other. There are several middle grounds between these two models, which may also be of our interest. An example is in the binary model, $F\left(\cdot\right)$ is known but the valuation function is non-parametric, i.e., $g\left(x_t\right)$ as in our model, or even $F\left(\cdot\right)$ is unknown and the valuation function is non-parametric.
>
> [F]Jianqing Fan, Yongyi Guo, Mengxin Yu (2021), Policy Optimization Using Semi-parametric Models for Dynamic Pricing, https://arxiv.org/abs/2109.06368.
>
> [G]Jianyu Xu, Yu-Xiang Wang (2022), Towards Agnostic Feature-based Dynamic Pricing: Linear Policies vs Linear Valuation with Unknown Noise, AISTATS 2022.
>
> [H] Luo, Y., Sun, W. W. (2021). Distribution-free contextual dynamic pricing. arXiv preprint arXiv:2109.07340.
>
> 4. Mismatching of the dimension $d$ in regret bound.
>
> We'd thank the reviewer for this suggestion. In the revised version, we will be more careful when we talk about the matching of lower bound and upper bound.

---

### Meta-Review · Area_Chair_wyKf · 2022-08-26

**Recommendation:** Accept
**Confidence:** Certain

**Metareview:**

The reviewers found the paper to be novel and interesting. The introduced model was found to be innovative and leading to cleaner/better regret bounds.  The only major concern that was raised and not resolved was lack of technical novelty. However, it seems that this work provides new and relevant results to the existing literature. And it seems that clean and fundamental techniques are indeed adequate for achieving the result of this paper.

**Award:**

No

---

### Decision · Program_Chairs · 2022-09-14

Accept